# Deep Generative Modeling on Limited Data with Regularization by Nontransferable Pre-trained Models

**Yong Zhong**[*12] **Hongtao Liu**[*12] **Xiaodong Liu**[12] **Fan Bao**[3] **Weiran Shen**[†12] **Chongxuan Li**[†12]

[1]Gaoling School of AI, Renmin University of China, Beijing, China
[2]Beijing Key Lab of Big Data Management and Analysis Methods, Beijing, China
[3]Department of Computer Science Technology, Tsinghua University, Beijing, China
`{yongzhong, ht6, xiaodong.liu}@ruc.edu.cn,`
`bf19@mails.tsinghua.edu.cn,`
`{shenweiran, chongxuanli}@ruc.edu.cn`

## Abstract

Deep generative models (DGMs) are data-eager because learning a complex model on limited data suffers from a large variance and easily overfits. Inspired by the classical perspective of the *bias-variance tradeoff*, we propose *regularized deep generative model* (Reg-DGM), which leverages a nontransferable pre-trained model to reduce the variance of generative modeling with limited data. Formally, Reg-DGM optimizes a weighted sum of a certain divergence and the expectation of an energy function, where the divergence is between the data and the model distributions, and the energy function is defined by the pre-trained model w.r.t. the model distribution. We analyze a simple yet representative Gaussian-fitting case to demonstrate how the weighting hyperparameter trades off the bias and the variance. Theoretically, we characterize the existence and the uniqueness of the global minimum of Reg-DGM in a non-parametric setting and prove its convergence with neural networks trained by gradient-based methods. Empirically, with various pre-trained feature extractors and a data-dependent energy function, Reg-DGM consistently improves the generation performance of strong DGMs with limited data and achieves competitive results to the state-of-the-art methods. Our implementation is available at `https://github.com/ML-GSAI/Reg-ADA-APA`.

## 1 Introduction

Deep generative models (DGMs) (Kingma & Welling, 2013; Goodfellow et al., 2014; Sohl-Dickstein et al., 2015; Van den Oord et al., 2016; Dinh et al., 2016; Hinton & Salakhutdinov, 2006) employ neural networks to capture the underlying distribution of high-dimensional data and find applications in various learning tasks (Kingma et al., 2014; Zhu et al., 2017; Razavi et al., 2019; Ramesh et al., 2021; 2022; Ho et al., 2022). Such models are often data-eager (Li et al., 2021; Wang et al., 2018) due to the presence of complex function classes. Recent work (Karras et al., 2020a) found that the classical variants of generative adversarial networks (GANs) (Goodfellow et al., 2014; Karras et al., 2020b) produce poor samples with limited data, which is shared by other DGMs in principle. Thus, improving the sample efficiency is a common challenge for DGMs.

The root cause of the problem is that learning a model in a complex class on limited data suffers from a large variance and easily overfits the training data (Mohri et al., 2018). To relieve the problem, previous work either employed sophisticated data augmentation strategies (Zhao et al., 2020a; Karras et al., 2020a; Jiang et al., 2021), or designed new losses for the discriminator in GANs (Cui et al., 2021; Yang et al., 2021), or transferred a pre-trained DGM (Wang et al., 2018; Noguchi & Harada, 2019; Mo et al., 2020). Although not pointed out in the literature to our knowledge, prior work can be understood as reducing the variance of the estimate implicitly (Mohri et al., 2018). In

---

[*]Equal contribution.
[†]Correspondence to Weiran Shen and Chongxuan Li.

this perspective, we propose a complementary framework, named *regularized deep generative model (Reg-DGM)*, which employs a pre-trained model as regularization to achieve a better *bias-variance tradeoff* when training a DGM with limited data.

In Sec. 2, we formulate the objective function of Reg-DGM as the sum of a certain divergence and a regularization term weighted by a hyperparameter. The divergence is between the data distribution and model distribution, and the regularization term can be understood as the negative expected log-likelihood of an energy-based model, whose energy function is defined by a pre-trained model, w.r.t. the model distribution. Intuitively, with an appropriate weighting hyperparameter, Reg-DGM balances between the data distribution and the pre-trained model to achieve a better bias-variance tradeoff with limited data, as validated by a simple yet prototypical Gaussian-fitting example.

In Sec. 3, we characterize the optimization behavior of Reg-DGM in both non-parametric and parametric settings under mild regularity conditions. On one hand, we prove the existence and the uniqueness of the global minimum of the regularized optimization problem with the Kullback–Leibler (KL) and Jensen–Shannon (JS) divergence in the non-parametric setting. On the other hand, we prove that, parameterized by a standard neural network architecture, Reg-DGM converges with a high probability to a global (or local) optimum trained by gradient-based methods.

In Sec. 4, we specify the components in Reg-DGM. We employ strong variants of GANs (Karras et al., 2020b;a; Jiang et al., 2021) as the base DGM for broader interests. We consider a *nontransferable* setting where the pre-trained model does not necessarily have to be a generative model. Indeed, we employ several feature extractors, which are trained for non-generative tasks such as image classification, representation learning, and face recognition. Notably, such models cannot be directly used in the fine-tuning approaches (Wang et al., 2018; Mo et al., 2020). With a comprehensive ablation study, we define our default energy function as the expected mean squared error between the features of the generated samples and the training data. Such an energy not only fits our theoretical analyses, but also results in consistent and significant improvements over baselines in all settings.

In Sec. 5, we present experiments on several benchmarks, including the FFHQ (Karras et al., 2019), LSUN CAT (Yu et al., 2015), and CIFAR-10 (Krizhevsky et al., 2009) datasets. We compare Reg-DGM with a large family of methods, including the base DGMs (Karras et al., 2020b;a; Jiang et al., 2021), the transfer-based approaches (Wang et al., 2018; Mo et al., 2020), the augmentation-based methods (Zhao et al., 2020a) and others (Cui et al., 2021; Yang et al., 2021). With a classifier pre-trained on ImageNet (Deng et al., 2009), or an image encoder pre-trained on the CLIP dataset (Radford et al., 2021), or a face recognizer pre-trained on VGGFace2 (Cao et al., 2018), Reg-DGM consistently improves strong DGMs under commonly used performance measures with limited data and achieves competitive results to the state-of-the-art methods. Our results demonstrate that Reg-DGM can achieve a good bias-variance tradeoff in practice, which supports our motivation.

In Sec. 6, we present the related work. In Sec. 7, we conclude the paper and discuss limitations.

## 2 METHOD

The goal of generative modeling is to learn a model distribution $p_g$ (implicitly or explicitly) from a training set $\mathcal{S} = \{x_i\}_{i=1}^m$ of size $m$ on a sample space $\mathcal{X}$. The elements in $\mathcal{S}$ are assumed to be drawn i.i.d. according to an unknown data distribution $p_d \in \mathcal{P}_\mathcal{X}$, where $\mathcal{P}_\mathcal{X}$ is the set of all valid distributions over $\mathcal{X}$. A general formulation for learning generative models is minimizing a certain statistical divergence $\mathbb{D}(\cdot||\cdot)$ between the two distributions as follows:

$$\min_{p_g \in \mathcal{H}} \mathbb{D}(p_d||p_g), \tag{1}$$

where $\mathcal{H} \subset \mathcal{P}_\mathcal{X}$ is the hypothesis class, for instance, a set of distributions defined by neural networks in a deep generative model (DGM). Notably, the divergence in Eq. (1) is estimated by the Monte Carlo method over the training set $\mathcal{S}$ and its solution has a small bias if not zero. However, learning a DGM with limited data is challenging because solving the problem in Eq. (1) with a small sample size $m$ essentially suffers from a large variance and probably overfits (Mohri et al., 2018).

Inspired by the classical perspective of the *bias-variance tradeoff*, we propose to leverage an external model $f$ pre-trained on a related and large dataset (e.g., ImageNet) as a data-dependent regularization to reduce the variance of training a DGM on limited data (e.g., CIFAR-10), which is

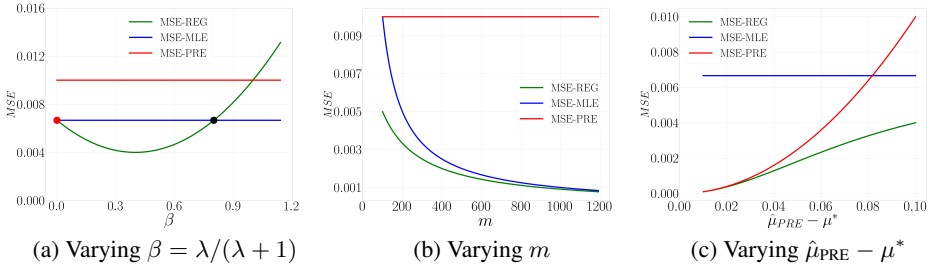

(a) Varying $\beta = \lambda/(\lambda+1)$      (b) Varying $m$      (c) Varying $\hat{\mu}_{\text{PRE}} - \mu^*$

Figure 1: MSE in the Gaussian-fitting example to validate Proposition 2.2. (a) The effect of $\lambda$ given a fixed $m$ and $\hat{\mu}_{\text{PRE}} - \mu^*$, where the red circle indicates $\beta = \max\left\{\frac{\sigma^2 - m(\hat{\mu}_{PRE}-\mu^*)^2}{\sigma^2 + m(\hat{\mu}_{PRE}-\mu^*)^2}, 0\right\}$ and the black circle indicates $\beta = \min\{\frac{2\sigma^2}{\sigma^2+m(\hat{\mu}_{PRE}-\mu^*)^2}, 1\}$. (b) The effect of the sample size $m$. (c) The effect of the bias of the pre-trained model. In both (b) and (c), the hyperparameter $\lambda$ is optimal.

complementary to prior work (see more details in Sec. 6). In particular, given a pre-trained $f$, we first introduce an energy function (LeCun et al., 2006) $\mathcal{E}_f : \mathcal{X} \to \mathbb{R}$ that satisfies mild regularity conditions (see Assumption A.1) and then define a probabilistic energy-based model (EBM) $p_f$ as $p_f(x) \propto \exp(-\mathcal{E}_f(x))$. We can treat $p_f$ as a special estimate of $p_d$ if $f$ is pre-trained on a dataset closely related to $p_d$. In this perspective, $p_f$ has a potentially large bias yet a variance of zero. To trade off the bias and variance, we simply optimize a weighted sum of the statistical divergence as in Eq. (1) and the expected log-likelihood of the EBM as follows:

$$\min_{p_g \in \mathcal{H}} \mathbb{D}(p_d||p_g) - \lambda \mathbb{E}_{x \sim p_g}[\log p_f(x)] \Leftrightarrow \min_{p_g \in \mathcal{H}} \mathbb{D}(p_d||p_g) + \lambda \mathbb{E}_{x \sim p_g}[\mathcal{E}_f(x)], \qquad (2)$$

where $\lambda > 0$ is the weighting hyperparameter balancing the two terms. The first one encourages $p_g$ to fit the data as in the original DGM and the second one encourages $p_g$ to produce samples with a high likelihood evaluated by the EBM $p_f$. Naturally, a properly selected $\lambda$ can hopefully achieve a better bias-variance tradeoff. We refer to our approach as *regularized deep generative model* (Reg-DGM) when $\mathcal{H}$ consists of distributions defined by neural networks.

In the following, we analyze a simple yet prototypical Gaussian-fitting example to demonstrate how a regularization term defined by a pre-trained model can relieve the bias-variance dilemma in generative modeling. Such an example is helpful to illustrate the motivation of Reg-DGM precisely and provide valuable insights on its practical performance. In the same spirit, representative prior work (Arjovsky et al., 2017; Mescheder et al., 2018) in the literature of DGMs has investigated similar examples with several parameters and closed-form solutions.

## 2.1 A PROTOTYPICAL GAUSSIAN-FITTING EXAMPLE

**Example 2.1** (Gaussian-fitting example). *The data distribution is a (univariate) Gaussian $p_d(x) = \mathcal{N}(x|\mu^*, \sigma^2)$, where $\sigma^2$ is known and $\mu^*$ is the parameter to be estimated. A training sample $\mathcal{S} = \{x_i\}_{i=1}^m$ is drawn i.i.d. according to $p_d(x)$. The hypothesis class for $p_g$ is $\mathcal{H} = \{\mathcal{N}(x|\mu, \sigma^2) \mid \mu \in \mathbb{R}\}$. The regularization term in Eq. (2) is $\mathcal{E}_f(x) := -\log \mathcal{N}(\hat{\mu}_{PRE}, \sigma^2)$, i.e., $p_f(x) = \mathcal{N}(x|\hat{\mu}_{PRE}, \sigma^2)$. Note that we let $\hat{\mu}_{PRE} \neq \mu^*$ and their gap $|\hat{\mu}_{PRE} - \mu^*|$ can be large in general.*

For simplicity, we consider the classical maximum likelihood estimation (MLE) (i.e., using the KL divergence in Eq. (1) as the performance measure), and its solution for Example 2.1 is given by the sample mean (Bishop & Nasrabadi, 2006): $\hat{\mu}_{\text{MLE}} = \frac{1}{m}\sum_{i=1}^m x_i, \hat{\mu}_{\text{MLE}} \sim \mathcal{N}\left(\mu^*, \frac{1}{m}\sigma^2\right)$. The pre-trained model $\hat{\mu}_{\text{PRE}}$ is another meaningful baseline for our approach. Clearly, it has a bias of $\hat{\mu}_{\text{PRE}} - \mu^*$ and a zero variance. Formally, the solution of our approach based on MLE is $\hat{\mu}_{\text{REG}} = \frac{1}{1+\lambda}\hat{\mu}_{\text{MLE}} + \frac{\lambda}{1+\lambda}\hat{\mu}_{\text{PRE}}, \hat{\mu}_{\text{REG}} \sim \mathcal{N}\left(\frac{1}{1+\lambda}\mu^* + \frac{\lambda}{1+\lambda}\hat{\mu}_{\text{PRE}}, \frac{\sigma^2}{m(1+\lambda)^2}\right)$. We compare all estimators under the mean squared error (MSE)[1], which is a common measure in statistics and machine learning. Formally, the MSE of an estimator $\hat{\theta}$ w.r.t. an unknown parameter $\theta$ is defined as:

---

[1]The expected risk coincides with the MSE in the example. See Appendix A.2.2 for generalization analyses.

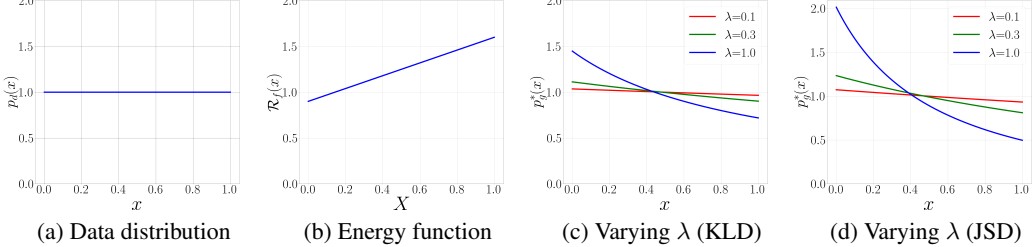

(a) Data distribution     (b) Energy function     (c) Varying $\lambda$ (KLD)     (d) Varying $\lambda$ (JSD)

Figure 2: Illustration of the optimization analyses. (a) The density of the data distribution. (b) The energy function defined by the pre-trained model. (c) The global minimum of the problem in Eq. (2) with the KL divergence (KLD) and different values of $\lambda$. (d) The global minimum of the problem in Eq. (2) with the JS divergence (JSD) and different values of $\lambda$. See more details in Appendix B.1.

$\mathrm{MSE}[\hat{\theta}] := \mathbb{E}[(\theta - \hat{\theta})^2]$, which can be decomposed as the sum of the variance of the estimator and the squared bias of the estimator (Bishop & Nasrabadi, 2006). As summarized in Proposition 2.2, our method achieves a better bias-variance tradeoff than the baselines if $\lambda$ is in an appropriate range.

**Proposition 2.2.** *Let* $\beta = \frac{\lambda}{\lambda+1}$ *be the normalized weight of the regularization term. In the Gaussian-fitting example 2.1, if* $\max\left\{\frac{\sigma^2 - m(\hat{\mu}_{PRE} - \mu^*)^2}{\sigma^2 + m(\hat{\mu}_{PRE} - \mu^*)^2}, 0\right\} < \beta < \min\left\{\frac{2\sigma^2}{\sigma^2 + m(\hat{\mu}_{PRE} - \mu^*)^2}, 1\right\}^2$, *then the following inequalities holds:*

$$MSE[\hat{\mu}_{REG}] < \min\{MSE[\hat{\mu}_{MLE}], MSE[\hat{\mu}_{PRE}]\}. \tag{3}$$

Due to space limit, the proof is deferred to Appendix A.1.1. We plot the MSE curves for all estimators w.r.t. the value of $\lambda$, $m$ and $|\hat{\mu_{\mathrm{PRE}}} - \mu^*|$ in Fig. 1 for a clearer illustration. Fig. 1 (a) directly validates the results of Proposition 2.2. Fig. 1 (b) and (c) show that with the optimal $\lambda$, the regularization gains more w.r.t. MLE as the sample size $m$ decreases and $|\hat{\mu_{\mathrm{PRE}}} - \mu^*|$ increases. See more details of the plot in Appendix B.1.

Since the generalization analysis in deep learning is still largely open (Belkin, 2021; Bartlett et al., 2021), it is difficult to generalize our Proposition 2.2 to the cases with deep models due to lack of appropriate tools. However, the intuition behind the Gaussian example also holds in deep learning. Namely, training a model on limited data suffers from a large variance (overfitting) and using a pretrained model suffers from a large bias (underfitting). Thus, Reg-DGM with a proper hyperparameter can balance between them and potentially achieve a better performance. Further, according to Fig. 1 (c), Reg-DGM benefits if the EBM defined by the pre-trained model is close to the target distribution, which inspires a data-dependent energy function presented in Sec. 4.

## 3 CONVERGENCE ANALYSES

### 3.1 ANALYSES IN THE NON-PARAMETRIC SETTING

We assume that our hypothesis class contains all valid distributions (i.e. $\mathcal{H} = \mathcal{P}_{\mathcal{X}}$), and the data distribution $p_d$ is accessible. Although the setting is impractical, such analyses characterize the existence and uniqueness of the global minimum in an ideal case and have been widely considered in deep generative models (Goodfellow et al., 2014; Arjovsky et al., 2017). Further, it is insightful to see how the regularization affects the solution of Reg-DGM in the ideal case.

Built upon the classical recipe of the calculus of variations and properties of the KL divergence in the topology of weak convergence, we establish our theory on the existence and uniqueness of the global minimum of Eq. (2) with the KL and JS divergence [3]. The results are formally characterized in Theorem 3.1 and Theorem 3.2 respectively. We refer the readers to Appendix A.2.1 for the

---

[2]Note that $\max\left\{\frac{\sigma^2 - m(\hat{\mu}_{\mathrm{PRE}} - \mu^*)^2}{\sigma^2 + m(\hat{\mu}_{\mathrm{PRE}} - \mu^*)^2}, 0\right\} < \min\left\{\frac{2\sigma^2}{\sigma^2 + m(\hat{\mu}_{\mathrm{PRE}} - \mu^*)^2}, 1\right\}$ always holds.

[3]We consider two divergences that are employed in the two representative DGMs: variational auto-encoders (VAE) (Kingma & Welling, 2013) and generative adversarial networks (GANs) (Goodfellow et al., 2014).

proofs. To establish our theory, we assume that (1) $\mathcal{X}$ is a nonempty compact set; (2) $\mathcal{E}_f : \mathcal{X} \to \mathbb{R}$ is continuous and bounded; (3) $\int_{\mathcal{X}} e^{-\mathcal{E}_f(x)} dx < \infty$, which are common and mild.

**Theorem 3.1.** *Under mild regularity conditions in Assumption A.1, for any $\lambda > 0$, there exists a unique global minimum of the problem in Eq. (2) with the KL divergence. Furthermore, the global minimum is in the form of $p_g^*(x) = \frac{p_d(x)}{\alpha^* + \lambda \mathcal{E}_f(x)}$, where $\alpha^* \in \mathbb{R}$.*

**Theorem 3.2.** *Under mild regularity conditions in Assumption A.1, for any $\lambda > 0$, there exists a unique global minimum of the problem in Eq. (2) with the JS divergence. Furthermore, the global minimum is in the form of $p_g^*(x) = \frac{p_d(x)}{e^{\alpha^* + \lambda \mathcal{E}_f(x)} - 1}$, where $\alpha^* \in \mathbb{R}$.*

As shown in Theorem 3.1 and Theorem 3.2, the global minimum is in the form of a reweighted data distribution and the weights are negatively correlated to the energy function defined by the pre-trained model. Qualitatively, the global minimum assigns high density for a sample $x$ if it has high density under the data distribution (i.e., $p_d(x)$) and low value of the energy function (i.e., $\mathcal{E}_f(x)$). Notably, the weights in Theorem 3.1 and Theorem 3.2 are different because of the different divergences. In particular, the effect of the pre-trained model is enlarged by the exponential term in Theorem 3.2 using JS divergence (JSD). Fig. 2 (c) and (d) show that with the same value of $\lambda$, the weighting coefficients of JSD are distributed in a larger range than KL divergence (KLD). Naturally, in both theorems, as $\lambda \to 0$, the denominator of $p_g^*(x)$ tends to a constant and $p_g^*(x) \to p_d(x)$, which recovers the solution of pure divergence minimization in Eq. (1). Therefore, Reg-DGM is consistent if the weighting parameter $\lambda$ is a function of $m$, and $\lim_{m \to \infty} \lambda(m) \to 0$.

## 3.2 ANALYSES IN THE PARAMETRIC SETTINGS

Although it provides theoretical insights of Reg-DGM, the non-parametric setting is far from practice. In fact, in our experiments, the hypothesis class is parameterized by neural networks and the training data is finite. In such a case, Reg-DGM can be formulated as a non-convex optimization problem, which is solved by gradient-based methods. Therefore, we also analyze the convergence of Reg-DGM trained by (stochastic) gradient descent in the presence of neural networks upon the general convergence framework (Allen-Zhu et al., 2019).

In particular, as summarized in Theorem 3.3, we show that Reg-DGM with a standard neural network architecture converges with a high probability under mild smoothness assumptions. The assumptions, result and proof are formally presented in Appendix A.2.2.

**Theorem 3.3** (Convergence of Reg-DGM (informal))**.** *Under standard and verifiable smoothness assumptions, with a high probability, Reg-DGM with a sufficiently wide ReLU CNN converges to a global optimum of Eq. (2) trained by GD and converges to a local minimum trained by SGD.*

## 4 IMPLEMENTATION

In this section, we discuss the base DGM, the pre-trained model and the energy function in practice.

### 4.1 BASE MODEL

Although Reg-DGM applies to variational auto-encoders (VAE) (Kingma & Welling, 2013) and many other DGMs, we focus on GANs (Goodfellow et al., 2014), which are most representative and popular in the scenarios with limited data (Karras et al., 2020a; Mo et al., 2020; Cui et al., 2021). Formally, GANs optimize an estimate of the JS divergence via a minimax formulation as follows:

$$\min_G \max_D \mathbb{E}_{x \sim p_d(x)}[\log D(x)] + \mathbb{E}_{x \sim p_g(x)}[\log(1 - D(x))], \tag{4}$$

where $G$ is a generator that defines $p_g(x)$ and $D$ is a discriminator that estimates the JS divergence by discriminating samples. Both $G$ and $D$ are parameterized by neural networks and Eq. (4) is estimated by the Monte Carlo method over mini-batches sampled from the training set.

For a broader interest, we adopt three strong GAN variants, StyleGAN2 (Karras et al., 2020b), adaptive discriminator augmentation (ADA) (Karras et al., 2020a), and adaptive pseudo augmentation (APA) (Jiang et al., 2021) as the base DGMs. Please refer to Appendix B.2 for more details.

## 4.2 Pre-trained Model

As mentioned in Sec. 2, different from the transfer-based methods (Wang et al., 2018; Mo et al., 2020), Reg-DGM applies to a *nontransferable* setting where the pre-trained model does not necessarily have the same architecture or the same formulation as $p_g$ or does not even have to be a generative model, enjoying the flexibility of choosing the pre-trained models.

In our implementation, the pre-trained model is a feature extractor $f : \mathcal{X} \to \mathbb{R}^d$, which is trained for other tasks (e.g., classification or contrastive representation learning) instead of generation. We choose such models because they are easily available and achieve excellent performance in supervised learning. In particular, we investigate three prototypical pre-trained models: a ResNet (He et al., 2015) trained in a supervised manner on ImageNet, a CLIP image encoder (Radford et al., 2021) trained in a self-supervised manner on a large-scale image-text dataset, and a FaceNet (Schroff et al., 2015) trained on a face recognition dataset. Please refer to Appendix B.2 for more details.

Note that such models are nontransferable in the fine-tuning manner (Hinton et al., 2006). Nevertheless, with such models and a data-dependent energy function presented later, Reg-DGM is still competitive to the transfer-based approaches (Wang et al., 2018; Mo et al., 2020) as shown in Tab. 1.

## 4.3 Energy Function

According to the results in Fig. 1 (c) and our intuition, we should define $\mathcal{E}_f$ such that $p_f$ is as close to $p_d$ as possible. In most of the cases, $f$ is pre-trained on a dataset with richer semantics than $p_d$. Therefore, it is necessary to involve training data (sampled from $p_d$) in the energy function to reducing the distance between $p_f$ and $p_d$. As presented above, we specify $f$ as a feature extractor and it is natural to match the features of samples from $p_g$ and $p_d$ as the energy function.

Formally, the energy function is defined by the expected mean squared error between the features of a generated sample and a training sample as follows:

$$\mathcal{E}_f(x) := \mathbb{E}_{x' \sim p_d} \left[ \frac{1}{d} ||f(x) - f(x')||_2^2 \right]. \tag{5}$$

Notably, our implementation with the data-dependent energy function in Eq. (5) is a valid instance of the general Reg-DGM framework as formulated in Eq. (2). Furthermore, the convergence results in both the non-parametric and parametric settings (see Sec. 3) hold in this case. The expectation is estimated by the Monte Carlo method of a single sample for efficiency by default and increasing the number of samples will not affect the performance significantly (see results in Appendix C.4).

We emphasize that our main contribution is not designing a specific energy function but the general framework of Reg-DGM. Many alternative energy functions can be employed in Reg-DGM. Indeed, we perform a systematical ablation study of the energy functions in Sec. 5.3 and find that Eq. (5) is the best among them considering the qualitative and quantitative results together. Moreover, we evaluate the effectiveness of Reg-DGM implemented by Eq. (5), with strong base DGMs (Karras et al., 2020b;a; Jiang et al., 2021), different datasets, different backbones of $f$ and different pre-training datasets in the experiments. We observe a consistent and significant improvement over SOTA baselines across various settings. Based on such a comprehensive empirical study, we believe that our implementation of Reg-DGM based on Eq. (5) would be effective in new settings.

## 5 Experiments

For a fair comparison to a large family of prior work, we evaluate Reg-DGM on several widely adopted benchmarks with limited data (Karras et al., 2020a), including the FFHQ (Karras et al., 2019), 100-shot Obama (Zhao et al., 2020a), LSUN CAT (Yu et al., 2015) and CIFAR-10 (Krizhevsky et al., 2009) datasets, and the data processing and metric calculating are the same as those of ADA (Karras et al., 2020a). We present the main results and analyses in the section and refer the readers to Appendix B.2 for details and Appendix C for additional results. Throughout the section, we refer to our approach as the name of the base DGM with the prefix "Reg-". For instance, "Reg-ADA" denotes our approach with ADA (Karras et al., 2020a) as the base DGM.

Table 1: Median FID ↓ on FFHQ and LSUN CAT and mean FID ↓ on CIFAR-10. $^{\dagger}$ and $^{\ddagger}$ indicate the results are taken from the references and Karras et al. (2020a) respectively. Otherwise, the results are reproduced by us upon the official implementation (Karras et al., 2020a; Jiang et al., 2021).

| Method | FFHQ | | LSUN CAT | | CIFAR-10 |
|---|---|---|---|---|---|
| | 1k | 5k | 1k | 5k | 50k |
| Transfer (Wang et al., 2018) | 21.42 | 12.34 | | | |
| Freeze-D (Mo et al., 2020) | 19.77 | 12.69 | | | |
| DA$^{\dagger}$ (Zhao et al., 2020a) | 25.66 | 10.45 | 42.26 | 16.11 | 8.49 |
| InsGen$^{\dagger}$ (Yang et al., 2021) | 19.58 | | | | |
| GenCo$^{\dagger}$ (Cui et al., 2021) | 65.31 | 27.96 | 140.08 | 40.79 | $8.83 \pm 0.04$ |
| DA + GenCo$^{\dagger}$ (Cui et al., 2021) | | | | | $6.57 \pm 0.01$ |
| ADA + bCR$^{\ddagger}$ (Zhao et al., 2020b) | 22.61 | 10.58 | 38.82 | 16.80 | |
| $R_{\text{LC}}$ $^{\dagger}$ (Tseng et al., 2021) | 63.16 | 23.83 | | | $8.31 \pm 0.05$ |
| ADA + $R_{\text{LC}}^{\dagger}$ (Tseng et al., 2021) | 21.7 | | | | $\mathbf{2.47 \pm 0.01}$ |
| APA$^{\dagger}$ (Jiang et al., 2021) | 45.19 | 13.25 | | | |
| StyleGAN2 (Karras et al., 2020b) | 103.66 | 52.71 | 186.55 | 115.16 | $7.16 \pm 0.12$ |
| Reg-StyleGAN2 (**ours**) | 75.99 | 37.77 | 107.02 | 63.10 | $6.56 \pm 0.14$ |
| ADA (Karras et al., 2020a) | 22.26 | 12.64 | 41.81 | 16.76 | $3.07 \pm 0.08$ |
| Reg-ADA (**ours**) | 20.05 | 11.95 | 36.17 | 15.91 | $2.95 \pm 0.05$ |
| ADA + APA (Jiang et al., 2021) | 19.71 | 8.84 | 24.09 | 11.79 | $2.64 \pm 0.08$ |
| Reg-ADA-APA (**ours**) | **17.88** | **8.02** | **21.88** | **11.27** | $2.58 \pm 0.04$ |

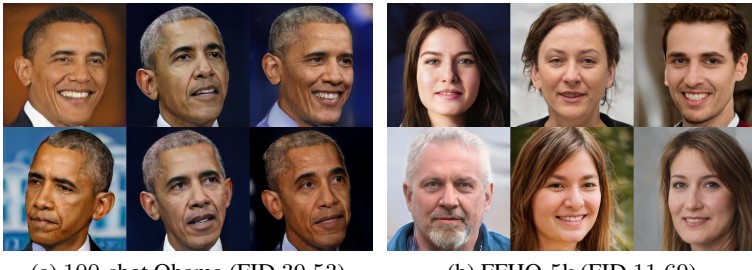

(a) 100-shot Obama (FID 39.53)  (b) FFHQ-5k (FID 11.69)

Figure 3: Samples from the Reg-ADA, truncated ($\psi = 0.7$) as in prior work (Karras et al., 2020a).

## 5.1 BENCHMARK RESULTS WITH LIMITED DATA

We employ StyleGAN2 (Karras et al., 2020b), ADA (Karras et al., 2020a) and APA (Jiang et al., 2021) as the base DGMs and a ResNet-18 (He et al., 2015) classifier trained on ImageNet (Deng et al., 2009) as the pre-trained model by default. The associated energy function is defined in Eq. (5).

Quantitatively, we compare Reg-DGM with a large family of existing methods, including the base DGMs, the transfer-based approaches, the augmentation-based methods and many others. Following the direct and strong competitor (Karras et al., 2020a), we report the median Fréchet inception distance (FID) (Heusel et al., 2017) on FFHQ and LSUN CAT, and the mean FID on CIFAR-10 out of 3 runs for a fair comparison in Tab. 1. For completeness, we also report the mean FID with the standard deviation on FFHQ and LSUN CAT in Appendix C.1.

As shown in Tab. 1, Reg-StyleGAN2, Reg-ADA and Reg-ADA-APA consistently outperform the corresponding base DGM in five settings, demonstrating that Reg-DGM can achieve a good bias-variance tradeoff in practice. Besides, the superior performance of Reg-ADA over ADA (and Reg-ADA-APA over ADA-APA) shows that our contribution is orthogonal to the augmentation-based approaches. Notably, the improvement of Reg-DGM over the base DGM is larger when the sample size $m$ is smaller. This is as expected because the relative gain of Reg-DGM over the base DGM

Table 2: Median FID ↓ and the corresponding KID$\times 10^3$ ↓ using a pre-trained CLIP or FaceNet.

| Method | CLIP | | | | FaceNet | |
| | FFHQ-5k | | LSUN CAT-5k | | FFHQ-5k | |
| | FID | KID | FID | KID | FID | KID |
| --- | --- | --- | --- | --- | --- | --- |
| StyleGAN2 (Karras et al., 2020b) | 52.71 | 39.52 | 115.16 | 100.57 | 52.71 | 39.52 |
| Reg-StyleGAN2(**ours**) | 40.98 | 27.56 | 42.04 | 26.21 | 38.80 | 23.38 |
| ADA (Karras et al., 2020a) | 12.64 | 5.17 | 16.76 | 8.13 | 12.64 | 5.17 |
| Reg-ADA(**ours**) | 11.09 | 3.91 | 14.15 | 6.72 | 11.37 | 4.01 |
| ADA+APA (Jiang et al., 2021) | 8.84 | 2.76 | 11.79 | 4.86 | 8.84 | 2.76 |
| Reg-ADA-APA(**ours**) | **8.18** | **2.26** | **10.47** | **4.68** | **8.21** | **2.37** |

increases as $m$ decreases given a fixed pre-trained model and the optimal $\lambda$ (see Fig. 1 (b)). Further, we evaluate the base DGMs and Reg-DGMs under the kernel inception distance (KID) (Bińkowski et al., 2018) metric in Appendix C.4. The conclusion remains the same. As suggested by Proposition 2.2, the value of the weighting hyperparameter $\lambda$ is crucial and there is an appropriate range of $\lambda$ such that Reg-DGM is better than the base DGM. We empirically validate the argument on FFHQ-5k with StyleGAN2 as the base model in Appendix C.5.

We mention that the methods based on fine-tuning (Wang et al., 2018; Mo et al., 2020) in Tab. 1 employ a GAN pre-trained on CelebA-HQ (Karras et al., 2017), which is also a face dataset of the same resolution as FFHQ. In comparison, our approach is built upon a classifier pertained on ImageNet. As highlighted in (Wang et al., 2018), the density of the pre-training dataset is more important than the diversity. Nevertheless, Reg-DGM is competitive with these strong baselines while enjoying the flexibility of choosing the pre-trained model and dataset. Further, we directly adopt the same pre-trained model across all datasets including CIFAR-10, where it takes additional efforts to get a suitable pre-trained generative model to fine-tune. Based on the results, it is safe to emphasize the complementary role of Reg-DGM to the approaches based on fine-tuning.

Qualitatively, we show the samples generated from Reg-DGM on FFHQ-5k and 100-shot Obama in Fig. 3. It can be seen that with the regularization, our approach can produce faces of a normal shape with limited data. We present the results from the base DGMs and more samples in other settings in Appendix C.2. For a comprehensive comparison on the visual quality of samples, we perform a human evaluation by the Amazon Mechanical Turk (AMT) as in prior work (Choi et al., 2020). According with the FID results, Reg-StyleGAN2 trained on FFHQ-5k against the baseline StyleGAN2 is chosen in 63.5% of the 3,000 image quality comparison tasks. See details in Appendix C.4.

## 5.2 ABLATION OF PRE-TRAINED MODELS AND PRE-TRAINING DATASETS

For simplicity, the main results of Reg-DGM presented in Sec. 5.1 are based on a ResNet-18 model pre-trained on ImageNet. Although its architecture is significantly different from the Inception-v3 (Szegedy et al., 2016) used in FID and KID calculation, it is worth performing an ablation on the pre-trained models to eliminate the potential bias caused by the pre-training dataset.

In particular, we test Reg-DGM with the image encoder of the CLIP model (Radford et al., 2021) (an architecture very similar to ResNet-50), which is pre-trained on large-scale noisy text-image pairs instead of ImageNet, and with the face recognizer Inception-ResNet-v1 (Szegedy et al., 2017) of FaceNet (Schroff et al., 2015), which is pre-trained on the large-scale face dataset VGGFace2 (Cao et al., 2018). See details in Appendix B.2. The median FID and corresponding KID results are shown in Tab. 2. Without heavily tuning the hyperparameter $\lambda$, Reg-DGM shows consistent improvements over the two baselines under both FID and KID metrics, and achieves comparable results to those presented in Table 1. The results with the CLIP model and the FaceNet model demonstrate that Reg-DGM works well with various backbones and pre-training datasets.

### 5.3 ABLATION OF ENERGY FUNCTIONS

As emphasized in Sec. 4, our main contribution is the general framework instead of a specific energy function. Nevertheless, we still investigate two alternative regularization terms for completeness.

We first consider the famous entropy-minimization regularization (Grandvalet & Bengio, 2004), which is data-independent. Intuitively, the regularization forces $p_g$ that produces samples with clear semantics. We find that the entropy regularization achieves similar FID results of $50.87$ on FFHQ-5k to the baseline $52.71$, showing the importance of the data dependency in the energy function. We then investigate another data-dependent regularization term, i.e., feature matching (Salimans et al., 2016), which matches the averaged features between the model and data distributions. We find that feature matching can achieve FID $32.65$ on FFHQ-5k greatly reducing the FID of StyleGAN2 while it cannot improve the visual quality of the samples. Please refer to Appendix C.4 for details of both energy functions. Therefore, Eq. (5) is the best among them considering the qualitative and quantitative results together and we believe it can be transferred to new settings based on our results in Sec. 5.1 and Sec. 5.2.

## 6 RELATED WORK

**Fine-tuning approaches.** A milestone of deep learning is that a deep generative model fine-tuned for classification outperforms the classical SVM on recognizing the hand-writing digits (Hinton et al., 2006). Since then, the idea of fine-tuning has a significant impact (Devlin et al., 2018; He et al., 2020) including generative models with limited data (Wang et al., 2018; Mo et al., 2020; Wang et al., 2020; Li et al., 2020; Ojha et al., 2021). However, an inherent restriction of fine-tuning is that the pre-trained model and target model should partially share a common structure. Thus, it may take additional efforts to find a suitable pre-trained model to fine-tune. In comparison, Reg-DGM provides an alternative way to make it possible to exploit a pre-trained classifier to help generative modeling. Notably, the latter is often thought of as much harder than the former.

**Other generative adversarial networks with limited data.** To relieve the overfitting problem of the discriminator, DA (Zhao et al., 2020a), ADA (Karras et al., 2020a) and APA (Jiang et al., 2021) design sophisticated data augmentation strategies. GenCo (Cui et al., 2021) designs a co-training framework that introduces multiple complimentary discriminators. InsGen (Yang et al., 2021) improves the data efficiency of GANs via an instance discrimination loss (Wu et al., 2018). We believe that Reg-DGM is orthogonal to these methods based on our results in Tab. 1.

**Regularization in probabilistic models.** Extensive regularization approaches have been developed in traditional Bayesian inference (Zhu et al., 2014) and probabilistic modeling (Chang et al., 2007; Liang et al., 2009; Ganchev et al., 2010). Among them, posterior regularization (PR) (Ganchev et al., 2010) encodes the human knowledge about the task as linear constraints of the latent representations in generative models for better inference performance. Such methods have been extended to deep generative models (Hu et al., 2018; Du et al., 2018; Shu et al., 2018; Xu et al., 2019) for a similar reason. Technically, PR-based methods regularize the latent space via handcrafted or jointly trained constraints. In comparison, our approach regularizes the data space via a pre-trained model. Besides, PR-based methods are suitable for structured prediction tasks instead of generative modeling with limited data, which is the main focus of this paper.

## 7 CONCLUSIONS

In this paper, we propose regularized deep generative model (Reg-DGM), which leverages a pre-trained model for regularization to reduce the variance of DGMs with limited data. Theoretically, we analyze the convergence properties of Reg-DGM. Empirically, with various pre-trained feature extractors and a data-dependent energy function, Reg-DGM consistently improves the generation performance of strong DGMs and achieves competitive results to the state-of-the-art methods. An interesting future work is to analyze the generalization behaviour of Reg-DGM in general and in-spire new energy functions. Currently, the generalization analysis of deep learning is still largely open (Zhang et al., 2021; Bartlett et al., 2021; Belkin, 2021), and there lacks appropriate tools to formalize our intuition on the bias-variance tradeoff in general.

## ACKNOWLEDGEMENT

We thank Guoqiang Wu for helpful discussions about the generalization analysis. This work was supported by NSF of China (Nos. 62076145, 62106273); Beijing Outstanding Young Scientist Program (No. BJJWZYJH012019100020098); Major Innovation & Planning Interdisciplinary Platform for the "Double-First Class" Initiative, Renmin University of China; the Fundamental Research Funds for the Central Universities, and the Research Funds of Renmin University of China (22XNKJ13, 22XNKJ16). Part of the computing resources supporting this work, totaled 720 A100 GPU hours, were provided by High-Flyer AI. (Hangzhou High-Flyer AI Fundamental Research Co., Ltd.). C. Li was also sponsored by Beijing Nova Program.

## ETHICS STATEMENT

This work presents a framework to train deep generative models on small data. By improving the data efficiency, it can potentially benefit real-world applications like medicine analysis and automatic drive. However, this work can have negative consequences in the form of "DeepFakes", as existing GANs. It is worth noting that this work may exacerbate such issues by improving the data efficiency of GANs. How to detect "DeepFakes" is an active research area in machine learning, which aims to relieve the problem.

## REPRODUCIBILITY STATEMENT

We submit the source code in the supplementary material and have released it. Datasets used in experiments are open and publicly available, and experimental details are provided in the Appendix B. In addition, the complete proof of the propositions and theorems is contained in the Appendix A.

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

## A  PROOFS

### A.1  THE GAUSSIAN-FITTING EXAMPLE

We first derive the solution of Reg-DGM in the main text. In the Gaussian fitting example, the regularized optimization problem can be written as

$$\hat{\mu}_{\text{REG}} = \arg\min_{\mu} -\frac{1}{m}\sum_{i=1}^{m}\log\mathcal{N}(\mu,\sigma^2) + \lambda\mathbb{E}_{x\sim\mathcal{N}(\mu,\sigma^2)}\left[-\log p_f(x)\right] \tag{6}$$

$$= \arg\min_{\mu}\frac{1}{m}\sum_{i=1}^{m}\frac{(\mu-x_i)^2}{2\sigma^2} + \lambda\left(\frac{(\mu-\hat{\mu}_{\text{PRE}})^2}{2\sigma^2} + \frac{1}{2}\log(2\pi\sigma^2) + \frac{1}{2}\right) \tag{7}$$

$$= \arg\min_{\mu}\frac{1}{m}\sum_{i=1}^{m}\frac{(\mu-x_i)^2}{2\sigma^2} + \lambda\frac{(\mu-\hat{\mu}_{\text{PRE}})^2}{2\sigma^2}, \tag{8}$$

where the first equality holds by the definition and properties of Gaussian (Bishop & Nasrabadi, 2006) and the second equality holds by omitting a constant irrelevant to the optimization. It is easy to solve the above quadratic programming problem analytically:

$$\hat{\mu}_{\text{REG}} = \frac{1}{m(1+\lambda)}\sum_{i=1}^{m}x_i + \frac{\lambda}{1+\lambda}\hat{\mu}_{\text{PRE}} = \frac{1}{1+\lambda}\hat{\mu}_{\text{MLE}} + \frac{\lambda}{1+\lambda}\hat{\mu}_{\text{PRE}}, \tag{9}$$

where $\hat{\mu}_{\text{MLE}} \sim \mathcal{N}\left(\mu^*, \frac{\sigma^2}{m}\right)$. $\hat{\mu}_{REG}$ is obtained by an affine transformation of a Gaussian random variable and thus is also Gaussian distributed as follows:

$$\hat{\mu}_{\text{REG}} \sim \mathcal{N}\left(\frac{1}{1+\lambda}\mu^* + \frac{\lambda}{1+\lambda}\hat{\mu}_{\text{PRE}}, \frac{\sigma^2}{m(1+\lambda)^2}\right). \tag{10}$$

#### A.1.1  PROOF OF PROPOSITION 2.2

*Proof.* In the Gaussian-fitting example, we have

$$\hat{\mu}_{\text{MLE}} \sim \mathcal{N}\left(\mu^*, \frac{\sigma^2}{m}\right), \tag{11}$$

and

$$\hat{\mu}_{\text{REG}} \sim \mathcal{N}\left(\frac{1}{1+\lambda}\mu^* + \frac{\lambda}{1+\lambda}\hat{\mu}_{\text{PRE}}, \frac{\sigma^2}{m(1+\lambda)^2}\right). \tag{12}$$

According to the bias-variance decomposition of the MSE, we have

$$\text{MSE}[\hat{\mu}_{\text{MLE}}] = \frac{\sigma^2}{m}, \tag{13}$$

and

$$\text{MSE}[\hat{\mu}_{\text{REG}}] = \frac{\lambda^2}{(1+\lambda)^2}(\hat{\mu}_{\text{PRE}} - \mu^*)^2 + \frac{\sigma^2}{m(1+\lambda)^2}. \tag{14}$$

Let $\beta = \frac{\lambda}{1+\lambda} \in (0,1)$ be the normalized weight of the regularization term. Then, we can rewrite $\text{MSE}[\hat{\mu}_{\text{REG}}]$ as

$$\text{MSE}[\hat{\mu}_{\text{REG}}] = \beta^2(\hat{\mu}_{\text{PRE}} - \mu^*)^2 + (1-\beta)^2\frac{\sigma^2}{m}. \tag{15}$$

To satisfy $\text{MSE}[\hat{\mu}_{\text{REG}}] < \text{MSE}[\hat{\mu}_{\text{MLE}}]$, we have

$$\beta^2(\hat{\mu}_{\text{PRE}} - \mu^*)^2 + (1-\beta)^2\frac{\sigma^2}{m} < \frac{\sigma^2}{m} \Rightarrow \beta < \frac{2\sigma^2}{\sigma^2 + m(\hat{\mu}_{\text{PRE}} - \mu^*)^2}. \tag{16}$$

The pre-trained model $\hat{\mu}_{\text{PRE}}$ is another meaningful baseline, which has a bias of $\hat{\mu}_{\text{PRE}} - \mu^*$ and a zero variance. Its MSE is given by

$$\text{MSE}[\hat{\mu}_{\text{PRE}}] = (\hat{\mu}_{\text{PRE}} - \mu^*)^2. \tag{17}$$

To satisfy $\text{MSE}[\hat{\mu}_{\text{REG}}] < \text{MSE}[\hat{\mu}_{\text{PRE}}]$, we have

$$\beta^2(\hat{\mu}_{\text{PRE}} - \mu^*)^2 + (1-\beta)^2\frac{\sigma^2}{m} < (\hat{\mu}_{\text{PRE}} - \mu^*)^2 \Rightarrow \beta > \frac{\sigma^2 - m(\hat{\mu}_{\text{PRE}} - \mu^*)^2}{\sigma^2 + m(\hat{\mu}_{\text{PRE}} - \mu^*)^2}, \tag{18}$$

which completes the proof. □

We now computes the optimal $\text{MSE}[\hat{\mu}_{\text{REG}}]$. It is easy to see that the quadratic programming problem in Eq. (15) achieves its minimum at $\beta^* = \frac{\sigma^2}{\sigma^2+m(\hat{\mu}_{\text{PRE}}-\mu^*)^2}$ with the corresponding $\lambda^* = \frac{\sigma^2}{m(\hat{\mu}_{\text{PRE}}-\mu^*)^2}$. The minimum value is $\frac{\sigma^2(\hat{\mu}_{\text{PRE}}-\mu^*)^2}{\sigma^2+m(\hat{\mu}_{\text{PRE}}-\mu^*)^2} = \frac{\text{MSE}[\hat{\mu}_{\text{MLE}}]\text{MSE}[\hat{\mu}_{\text{PRE}}]}{\text{MSE}[\hat{\mu}_{\text{MLE}}]+\text{MSE}[\hat{\mu}_{\text{PRE}}]}$.

### A.1.2 COMPARISON UNDER THE EXPECTED RISK

We also evaluate all estimators in terms of the *expected risk*, which is a commonly used measure in statistical learning theory. In a density estimation task for $p_d$, the expected risk of a hypothesis $\hat{\mu}$, which depends on the training sample $\mathcal{S}$, is $R(\hat{\mu}) := \mathbb{E}_{x \sim p_d}[-\log p(x; \hat{\mu})]$. We can show that the expectation of the expected risk w.r.t. $\mathcal{S}$ coincides with the corresponding MSE in the Gaussian-fitting example and directly obtain the following Corollary A.0.1 from Proposition 2.2.

**Corollary A.0.1.** *In the Gaussian-fitting example 2.1, if $\lambda$ satisfies the same condition as in Proposition 2.2, then the following inequality holds:*

$$\mathbb{E}_{\mathcal{S}}[R(\hat{\mu}_{REG})] < \min\{\mathbb{E}_{\mathcal{S}}[R(\hat{\mu}_{MLE})], \mathbb{E}_{\mathcal{S}}[R(\hat{\mu}_{PRE})]\}. \tag{19}$$

*Proof.* In the Gaussian-fitting example, the expected risk for a hypothesis $\hat{\mu}$ is

$$R(\hat{\mu}) = \mathbb{E}_{\mathcal{S} \sim p_d^m}\mathbb{E}_{x \sim \mathcal{N}(\mu^*, \sigma^2)}\left[-\log\mathcal{N}(\hat{\mu}, \sigma^2)\right] \tag{20}$$

$$= \mathbb{E}_{\mathcal{S} \sim p_d^m}\left[\frac{(\hat{\mu} - \mu^*)^2}{2\sigma^2} + \frac{1}{2}\log(2\pi\sigma^2) + \frac{1}{2}\right] \tag{21}$$

$$= \frac{1}{2\sigma^2}\text{MSE}[\hat{\mu}] + \frac{1}{2}\log(2\pi\sigma^2) + \frac{1}{2}, \tag{22}$$

which completes the proof together with Proposition 2.2. □

### A.2 OPTIMIZATION ANALYSES

### A.2.1 PROOF OF THEOREM 3.1 AND THEOREM 3.2

Our results rely on the following regularity conditions.

**Assumption A.1.**

1. $\mathcal{X}$ *is a nonempty compact set.*

2. $\mathcal{E}_f : \mathcal{X} \to \mathbb{R}$ *is continuous and bounded.*

3. $\int_{\mathcal{X}} e^{-\mathcal{E}_f(x)} dx < \infty.$

The regularity conditions in Assumption A.1 are mild in the sense:

1. The sample space $\mathcal{X}$ is often a subset of a $n$-dimensional Euclidean space. Then, $\mathcal{X}$ is compact if and only if $\mathcal{X}$ is bounded and closed, which holds for extensive datasets in various types including images, videos, audios, and texts.

2. In this paper, $\mathcal{E}_f$ is defined by compositing a neural network and a continuous real-valued function. Then $\mathcal{E}_f$ is continuous and bounded on $\mathcal{X}$ if the neural network has bounded weights and uses continuous activation functions including ReLU (Nair & Hinton, 2010), Tanh and Sigmoid.

3. $\int_{\mathcal{X}} e^{-\mathcal{E}_f(x)} dx < \infty$ holds following the choice of the sample space and energy function,

We first prove Theorem 3.1 as follows.

*Proof.* Ignoring some constant irrelevant to the optimization, we rewrite the optimization problem of our approach with the KL divergence as follows:

$$
\min_{p_g} \int_{\mathcal{X}} (-\ln(p_g(x))p_d(x) + \lambda \mathcal{E}_f(x)p_g(x))dx,
$$
$$
\text{subject to } \int_{\mathcal{X}} p_g(x)dx = 1, \tag{23}
$$
$$
\forall x \in \mathcal{X}, p_g(x) \geq 0.
$$

For clarity, we denote the functional in problem Eq. (23) to be optimized as

$$
\mathcal{J}(p_g) := \int_{\mathcal{X}} (-\ln(p_g(x))p_d(x) + \lambda \mathcal{E}_f(x)p_g(x))dx.
$$

According to Assumption 3.1 that $\mathcal{X}$ is a nonempty compact set, the feasible area characterized by the constraints (i.e., $\mathcal{P}_{\mathcal{X}}$) is compact in the topology of weak convergence by the Prokhorov's Theorem (See Corollary 6.8 in (Van Gaans, 2003)). By Theorem 3.6 in (Ajjanagadde et al., 2017), KL divergence is lower semi-continuous in the topology of weak convergence. According to Assumption 3.2 that $\mathcal{E}_f$ is continuous and bounded on $\mathcal{X}$, the regularization term is continuous in the topology of weak convergence. Therefore, by the extreme value theorem, the global minimum of $\mathcal{J}(p_g)$ exists in the feasible area.

Note that the optimization problem Eq. (23) is convex. To obtain a necessary condition for the global minima, we get the Lagrangian with the equality constraint:

$$
\mathcal{L}(p_g) := \int_{\mathcal{X}} (-\ln(p_g(x))p_d(x) + \lambda \mathcal{E}_f(x)p_g(x))dx + \alpha(\int_{\mathcal{X}} p_g(x)dx - 1), \tag{24}
$$

where $\alpha \in \mathbb{R}$. Note that for simplicity, we do not include the inequality constraint, which will be verified shortly. It is easy to check the constraint qualifications for problem Eq. (23). By the calculus of variations, a necessary condition for a global minimum of problem Eq. (23) is

$$
\frac{\delta \mathcal{L}}{\delta p_g} = -\frac{p_d(x)}{p_g(x)} + \lambda \mathcal{E}_f(x) + \alpha = 0, \tag{25}
$$

which implies that

$$
p_g^*(x) = \frac{p_d(x)}{\alpha + \lambda \mathcal{E}_f(x)}. \tag{26}
$$

We define $\mathcal{A} = \{\alpha \in \mathbb{R} | \frac{p_d(x)}{\alpha + \lambda \mathcal{E}_f(x)} \geq 0, \forall x \in \mathcal{X}\}$. According to Assumption 3.1 that $\mathcal{E}_f$ is bounded on $\mathcal{X}$, we have $\mathcal{A} \neq \varnothing$. We define a function $\phi : \mathcal{A} \to \mathbb{R}$ as

$$\phi(\alpha) = \int_{\mathcal{X}} \frac{p_d(x)}{\alpha + \lambda \mathcal{E}_f(x)} dx. \tag{27}$$

It is easy to see $\phi(\alpha)$ is monotonically decreasing and there is at most one $\alpha^* \in \mathcal{A}$ such that $\phi(\alpha^*) = 1$, which finishes the proof together with the existence of the global minimum. $\quad\square$

We now prove Theorem 3.2 as follows.

*Proof.* The proof here shares the same spirit of Theorem 3.1. Ignoring some constant irrelevant to the optimization, we rewrite the optimization problem of our approach with the JS divergence as follows:

$$\min_{p_g} \int_{\mathcal{X}} (-\ln(p_g(x) + p_d(x))p_d(x) - \ln(p_g(x) + p_d(x))p_g(x) + \ln(p_g(x))p_g(x) + \lambda \mathcal{E}_f(x)p_g(x))dx,$$

$$\text{subject to } \int_{\mathcal{X}} p_g(x)dx = 1,$$

$$\forall x \in \mathcal{X}, p_g(x) \geq 0. \tag{28}$$

For clarity, we denote the functional in problem Eq. (23) to be optimized as

$$\mathcal{J}(p_g) := \int_{\mathcal{X}} (-\ln(p_g(x) + p_d(x))p_d(x) - \ln(p_g(x) + p_d(x))p_g(x) + \ln(p_g(x))p_g(x) + \tag{29}$$

$$\lambda \mathcal{E}_f(x)p_g(x))dx. \tag{30}$$

Similarly to the proof of Theorem 3.1, the global minimum of $\mathcal{J}(p_g)$ exists in the feasible area due to Assumption A.1 and the fact that the JS divergence is lower semi-continuous in the topology of weak convergence.

Notably, the optimization problem Eq. (28) is convex due to the convexity of the JS divergence (Billingsley, 2013). To obtain a necessary condition for the global minima, we get the Lagrangian with the equality constraint:

$$\mathcal{L}(p_g) := \int_{\mathcal{X}} (-\ln(p_g(x) + p_d(x))p_d(x) - \ln(p_g(x) + p_d(x))p_g(x) + \ln(p_g(x))p_g(x) +$$

$$\lambda \mathcal{E}_f(x)p_g(x))dx + \alpha(\int_{\mathcal{X}} p_g(x)dx - 1), \tag{31}$$

where $\alpha \in \mathbb{R}$. Similarly to the proof of Theorem 3.1, a necessary condition for a global minimum of problem Eq. (23) is

$$\frac{\delta \mathcal{L}}{\delta p_g} = -\ln(p_g(x) + p_d(x)) + \ln(p_g(x)) + \lambda \mathcal{E}_f(x) + \alpha = 0, \tag{32}$$

which implies that

$$p_g^*(x) = \frac{p_d(x)}{e^{\alpha + \lambda \mathcal{E}_f(x)} - 1}. \tag{33}$$

Similarly to the proof in Theorem 3.1, there is at most one $\alpha^* \in \mathbb{R}$ such that $\int_{\mathcal{X}} \frac{p_d(x)}{e^{\alpha + \lambda \mathcal{E}_f(x)} - 1} = 1$, which finishes the proof together with the existence of the global minimum. $\quad\square$

### A.2.2 CONVERGENCE WITH NEURAL NETWORKS TRAINED BY (STOCHASTIC) GRADIENT DESCENT

We establish the convergence of Reg-DGM with over-parameterized neural networks trained by (stochastic) gradient descent upon the general framework (Allen-Zhu et al., 2019).

**Theorem A.2.** *(General convergence guarantee (Allen-Zhu et al., 2019)) For an arbitrary Lipschitz-smooth loss function $\mathcal{L}$, with probability at least $1 - e^{-\Omega(\log m)}$, a ReLU convolutional neural network with width $m$ and depth $l$ trained by gradient descent with an appropriate learning rate satisfy the following.[4]*

- *If $\mathcal{L}$ is non-convex, and $\sigma$-gradient dominant, then GD finds $\epsilon$-error minimizer in $\tilde{O}(poly(n, l, \log \frac{1}{\epsilon}, \frac{1}{\sigma}))$ iterations as long as $m \geq \tilde{\Omega}(poly(n, l, \frac{1}{\sigma}))$.*

- *If $\mathcal{L}$ is non-convex, then SGD finds a point with $||\nabla f|| \leq \epsilon$ in $\tilde{O}(poly(m, l, \log \frac{1}{\epsilon}))$ iterations as long as $m \geq \tilde{\Omega}(poly(n, l, \frac{1}{\epsilon}))$.*

We assume the following standard smoothness conditions, which can be verified in practice with bounded data and weights.

**Assumption A.3.** *(Smoothness conditions)*

1. *$\exists b > 0$ such that $\forall \theta \in \Theta, \forall x \in \mathcal{X}, p(\theta; x) \geq b$.*

2. *$\exists L > 0$ such that $\forall x \in \mathcal{X}, \forall \theta \in \Theta, \forall \theta' \in \Theta, |p(\theta; x) - p(\theta'; x)| \leq L||\theta - \theta'||$.*

3. *$\exists K > 0$ such that $\forall \theta \in \Theta, \forall \theta' \in \Theta, \int |p(\theta; x) - p(\theta'; x)|dx \leq K||\theta - \theta'||$.*

4. *$\sup_{x \in \mathcal{X}} \sup_{y \in \mathcal{X}} ||f(x) - f(y)||^2 \leq B$.*

We consider the general density estimation setting where $\mathcal{L}_{\text{MLE}}(\theta; x_i) := -\log p_\theta(x_i)$. Note that the first two conditions in Assumption A.3 directly imply that $\mathcal{L}_{\text{MLE}}(\cdot; x)$ is $\frac{L}{b}$-Lipschitz. Formally, given a set of samples $\mathcal{S} = \{x_i\}_{i=1}^n$, Reg-DGM optimizes

$$\hat{R}[\theta] := \frac{1}{n} \sum_{i=1}^n \mathcal{L}_{\text{MLE}}(\theta; x_i) + \lambda \mathbb{E}_{x \sim p_g}[\mathcal{E}_f(x)].$$

If $\mathcal{E}_f$ is independent from the training data $x$, then the overall loss function is also $\frac{L}{b}$-Lipschitz and Theorem A.2 directly applies. Otherwise, we can show that the data-dependent regularization used in our experiments is also Lipschitz-smooth. By the linearity of expectation, we have

$$\hat{\theta}_{\text{REG}} := \arg\min_{\theta \in \Theta} \frac{1}{n} \sum_{i=1}^n [\mathcal{L}_{\text{MLE}}(\theta; x_i)] + \frac{\lambda}{d} \mathbb{E}_{y \sim p_\theta} \left[ \frac{1}{n} \sum_{i=1}^n ||f(y) - f(x_i)||_2^2 \right] \quad (34)$$

$$= \arg\min_{\theta \in \Theta} \frac{1}{n} \sum_{i=1}^n \left[ \mathcal{L}_{\text{MLE}}(\theta; x_i) + \frac{\lambda}{d} \mathbb{E}_{y \sim p_\theta} ||f(y) - f(x_i)||_2^2 \right]. \quad (35)$$

We define $\mathcal{L}_{\text{REG}}(\theta; x_i) := \frac{\lambda}{d} \mathbb{E}_{y \sim p_\theta} ||f(y) - f(x_i)||_2^2$, which is Lipschitz-smooth:

$$
\begin{aligned}
|\mathcal{L}_{\text{REG}}(\theta; x) - \mathcal{L}_{\text{REG}}(\theta'; x)| =& \frac{\lambda}{d} \left| \mathbb{E}_{y \sim p_\theta} ||f(y) - f(x)||_2^2 - \mathbb{E}_{y \sim p_{\theta'}} ||f(y) - f(x)||_2^2 \right| \\
\leq& \frac{\lambda}{d} \int |p_\theta(y) - p_{\theta'}(y)| ||f(x) - f(y)||^2 dy \\
\leq& \frac{\lambda}{d} (\sup_{y \in \mathcal{X}} ||f(x) - f(y)||^2) \int |p_\theta(y) - p_{\theta'}(y)| dy \\
\leq& \frac{\lambda BK}{d} ||\theta - \theta'||.
\end{aligned}
$$

Therefore, Theorem A.2 applies to Reg-DGM with the data-dependent energy defined in Sec. 4 of the main text.

## B EXPERIMENTAL DETAILS

Our implementation is built upon some publicly available code. Below, we include the links and please refer to the licenses therein.

---

[4]See additional assumptions on regularity of the data in (Allen-Zhu et al., 2019).

## B.1 TOY DATA

In the toy example for optimization analyses, the data follows a uniform distribution over $[0, 1]$. The energy function is defined as $\mathcal{E}_f(x) = 0.7x + 0.9$. The optimal $\beta^*$ is estimated by numerical integration.

In the experiments for the Gaussian-fitting example, we set $\sigma^2 = 1$, $m = 150$, and $\hat{\mu}_{\text{PRE}} - \mu^* = 0.1$ by default.

## B.2 GANs WITH LIMITED DATA

**Datasets.** In our experiments, we use FFHQ (Karras et al., 2019), which consists of $70,000$ human face images of resolution $256 \times 256$, LSUN CAT (Yu et al., 2015), which consists of $200,000$ cat images of resolution $256 \times 256$, and CIFAR-10 (Krizhevsky et al., 2009), which consists of $50,000$ natural images of resolution $32 \times 32$. Specifically, we randomly split training subsets of size 1k and 5k from full FFHQ and LSUN CAT in the same way as ADA (Karras et al., 2020a). For reproducibility, we directly use the random seed provided by the official implementation of ADA [5]. In addition, we also experiment on the smaller dataset 100-shot Obama (Zhao et al., 2020a) with only 100 face images of $256 \times 256$ resolution. In all experiments, we do not use x-flips to amplify training data except for combining with APA (Jiang et al., 2021).

**Metrics.** To quantitatively evaluate the experimental results, we choose the Fréchet inception distance (FID) (Heusel et al., 2017) and the kernel inception distance (KID) (Bińkowski et al., 2018) as our metrics. We compute the FID and KID between $50,000$ generated images and all real images instead of training subsets (Heusel et al., 2017). Following ADA (Karras et al., 2020a), we report the medium FID and corresponding KID on FFHQ and LSUN CAT and the mean FID with standard deviation on CIFAR-10 out of 3 runs. We record the best FID during training in each run as in ADA (Karras et al., 2020a).

**Base DGM.** In particular, the lighter-weight StyleGAN2 is the backbone for FFHQ and LSUN CAT, and the tuning StyleGAN2 is the backbone for CIFAR-10 following ADA (Karras et al., 2020a). Compared to the official StyleGAN2, the lighter-weight StyleGAN2 has the same performance and less computing cost on the FFHQ and LSUN CAT and the tuning StyleGAN2 is more suitable for CIFAR-10 (Karras et al., 2020a). Adaptive discriminator augmentation (ADA) (Karras et al., 2020a) is a representative way of data augmentation for GANs under limited data, and its combination with adaptive pseudo augmentation (APA) (Jiang et al., 2021) is the state-of-the-art method for few-shot image generation (Li et al., 2022). Our implementation is based on the official code of ADA [5] and APA [6].

**Pre-trained model.** We choose the ResNet-18 [7] trained on the ImageNet dataset as the pre-trained model by default due to its excellent performance on ImageNet (Deng et al., 2009) and little computational overhead. We normalize both the real and fake images based on the mean and standard deviation of training data and then feed them into the classifier. We extract the features of the last fully connected layer in the pre-trained model, which is frozen during training of generative models. On CIFAR-10, we interpolate [8] both the fake and real images to a resolution of $256 \times 256$ after normalization.

Other alternative pre-trained models are the image encoder ResNet-50 of the CLIP model (Radford et al., 2021), which is pre-trained on large-scale noisy text-image pairs instead of ImageNet, and the face recognizer Inception-ResNet-v1 (Szegedy et al., 2017) of FaceNet (Schroff et al., 2015), which is pre-trained on the large-scale face dataset VGGFace2 (Cao et al., 2018). Notably, we replace the last attention pooling layer in the image encoder of CLIP with the global average pooling layer, and we directly pass the image to the face recognizer without operating the face detector in FaceNet

---

[5] https://github.com/NVlabs/stylegan2-ada-pytorch
[6] https://github.com/EndlessSora/DeceiveD
[7] https://pytorch.org/vision/stable/models.html
[8] https://pytorch.org/docs/stable/generated/torch.nn.functional.interpolate.html

Table 3: Hyperparameters in the experiments of GANs. Reg-DGM shares the same hyperparameters as the base DGM if not specified. All models converge with the corresponding training length.

| Parameter | FFHQ-1k | FFHQ-5k | LSUN CAT-1k | LSUN CAT-5k | CIFAR-10 |
|---|---|---|---|---|---|
| Base DGM | StyleGAN2 (Karras et al., 2020b) | StyleGAN2 (Karras et al., 2020b) | StyleGAN2 (Karras et al., 2020b) | StyleGAN2 (Karras et al., 2020b) | StyleGAN2 (Karras et al., 2020b) |
| $\lambda$ | 4 | 1 | 4 | 1 | $1 \times 10^{-5}$ |
| Number of GPUs | 4 | 4 | 8 | 8 | 8 |
| Training length | 5M | 5M | 5M | 5M | 25M |
| Minibatch size | 64 | 64 | 64 | 64 | 64 |
| Base DGM | ADA (Karras et al., 2020a) | ADA (Karras et al., 2020a) | ADA (Karras et al., 2020a) | ADA (Karras et al., 2020a) | ADA (Karras et al., 2020a) |
| $\lambda$ | 0.01 | 0.001 | 0.01 | 0.0005 | $5 \times 10^{-6}$ |
| Number of GPUs | 8 | 8 | 8 | 8 | 8 |
| Training length | 16M | 16M | 16M | 16M | 60M |
| Minibatch size | 64 | 64 | 64 | 64 | 64 |
| Base DGM | ADA+APA (Jiang et al., 2021) | ADA+APA (Jiang et al., 2021) | ADA+APA (Jiang et al., 2021) | ADA+APA (Jiang et al., 2021) | ADA+APA (Jiang et al., 2021) |
| $\lambda$ | 0.01 | 0.01 | 0.005 | 0.001 | 0.1 |
| Number of GPUs | 8 | 8 | 8 | 8 | 8 |
| Training length | 25M | 25M | 25M | 25M | 100M |
| Minibatch size | 64 | 64 | 64 | 64 | 64 |

Table 4: The hyperparameter $\lambda$ with pre-trained CLIP and FaceNet.

| Base DGM | CLIP | | FaceNet |
|---|---|---|---|
| | FFHQ-5k | LSUN CAT-5k | FFHQ-5k |
| StyleGAN2 (Karras et al., 2020b) | 50 | 50 | 4 |
| ADA (Karras et al., 2020a) | 1 | 2 | 0.05 |
| ADA+APA (Jiang et al., 2021) | 0.5 | 1 | 0.001 |

to extract cropped faces. For pre-trained CLIP[9] and FaceNet[10], we also employ their last layers as feature extractors.

**Hyperparameters.** Some parameters are shown in Tab. 3. The weight parameter $\lambda$ controls the strength of our regularization term. We choose the weighting hyperparameter $\lambda$ by performing grid search over $[50, 20, 10, 5, 4, 2, 1, 0.5, 0.1, 0.05, 0.01, 0.005, 0.001, 0.0005]$ for FFHQ and LSUN CAT, and $[1, 0.1, 0.05, 0.01, 0.005, 0.001, 0.0005, 0.0001, 0.00005, 0.00001, 0.000005]$ for CIFAR-10 according to FID following prior work (Karras et al., 2020a). Other parameters remain the same settings as ADA (Karras et al., 2020a) and APA (Jiang et al., 2021). For pre-trained CLIP and FaceNet, the adopted parameter $\lambda$ is shown in Tab. 4. In addition, we simply set $\lambda$ as 0.1 for Reg-ADA trained on 100-shot Obama.

**Computing amount.** A single experiment can be completed on 8 NVIDIA 2080Ti GPUs. It takes 1 day 6 hours 19 minutes to run our method with ADA on FFHQ or LSUN CAT and 2 days 17 hours 10 minutes on CIFAR-10 at a time.

## C  ADDITIONAL RESULTS

### C.1  STANDARD DEVIATION ON FFHQ AND LSUN CAT

As shown in Tab. 5, we also provide the mean FID and standard deviation on FFHQ and LSUN CAT. Reg-DGM can reduce the mean FID significantly (compared to the measurement variance) and achieve a similar if not smaller standard deviation.

---

[9] https://github.com/openai/CLIP
[10] https://github.com/timesler/facenet-pytorch

Table 5: The mean FID ↓ and standard deviation on FFHQ and LSUN CAT which is a supplement to reported medium FID.

| Method | FFHQ | | LSUN CAT | |
|---|---|---|---|---|
| | 1k | 5k | 1k | 5k |
| StyleGAN2 (Karras et al., 2020b) | $102.62 \pm 5.67$ | $53.37 \pm 1.92$ | $189.57 \pm 8.13$ | $110.83 \pm 6.85$ |
| Reg-StyleGAN2 (**ours**) | $\mathbf{77.80 \pm 3.65}$ | $\mathbf{38.14 \pm 0.97}$ | $\mathbf{112.15 \pm 8.48}$ | $\mathbf{64.11 \pm 2.51}$ |
| ADA (Karras et al., 2020a) | $22.10 \pm 0.50$ | $12.72 \pm 0.13$ | $41.59 \pm 1.71$ | $16.77 \pm 0.74$ |
| Reg-ADA (**ours**) | $\mathbf{20.16 \pm 0.22}$ | $\mathbf{11.88 \pm 0.13}$ | $\mathbf{36.85 \pm 1.09}$ | $\mathbf{15.85 \pm 0.10}$ |

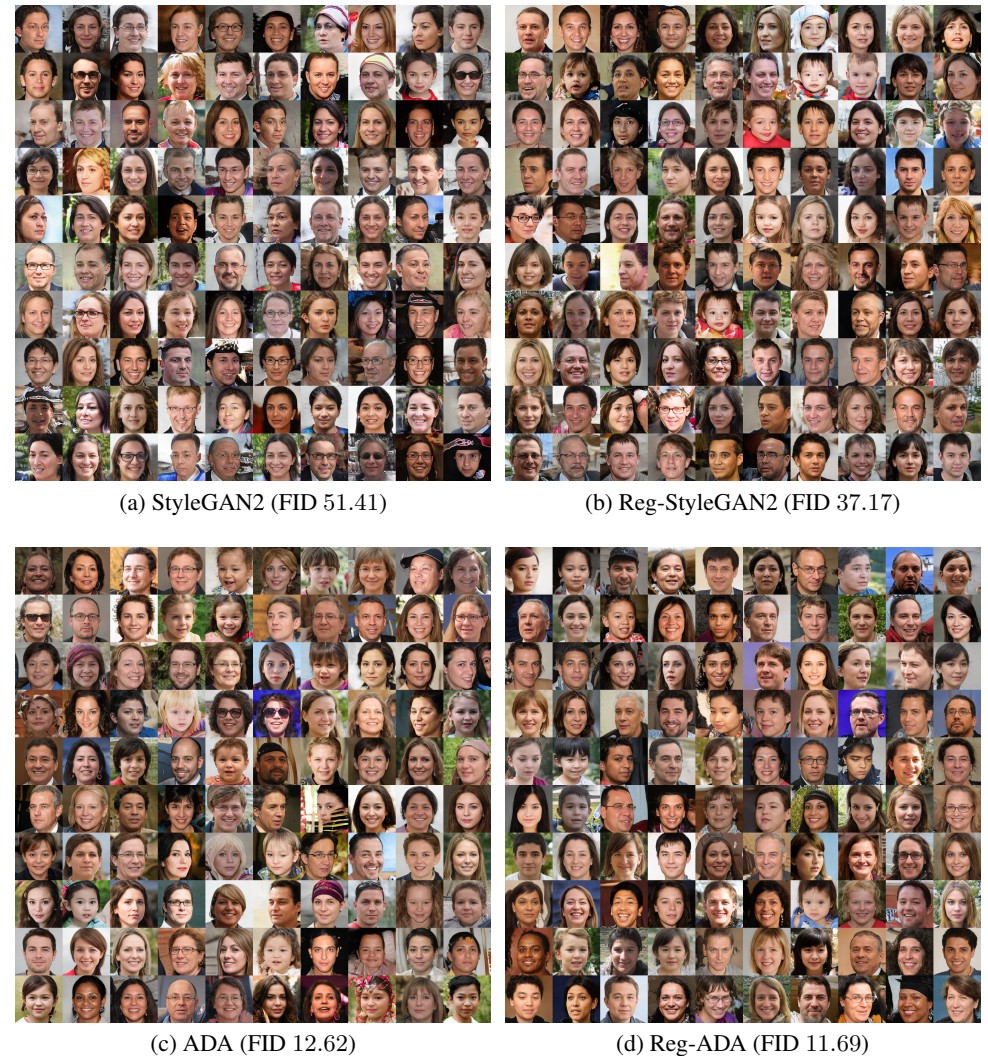

(a) StyleGAN2 (FID 51.41)  (b) Reg-StyleGAN2 (FID 37.17)

(c) ADA (FID 12.62)  (d) Reg-ADA (FID 11.69)

Figure 4: Samples generated for FFHQ-5k, truncated ($\psi = 0.7$).

## C.2 MORE SAMPLES OF GANS

Fig. 4 and Fig. 5 respectively show the samples randomly generated by models with best FID trained on FFHQ-5k and CIFAR-10, using slight truncation as in ADA (Karras et al., 2020a). Reg-DGM produces samples of better or comparable quality to the corresponding base DGM.

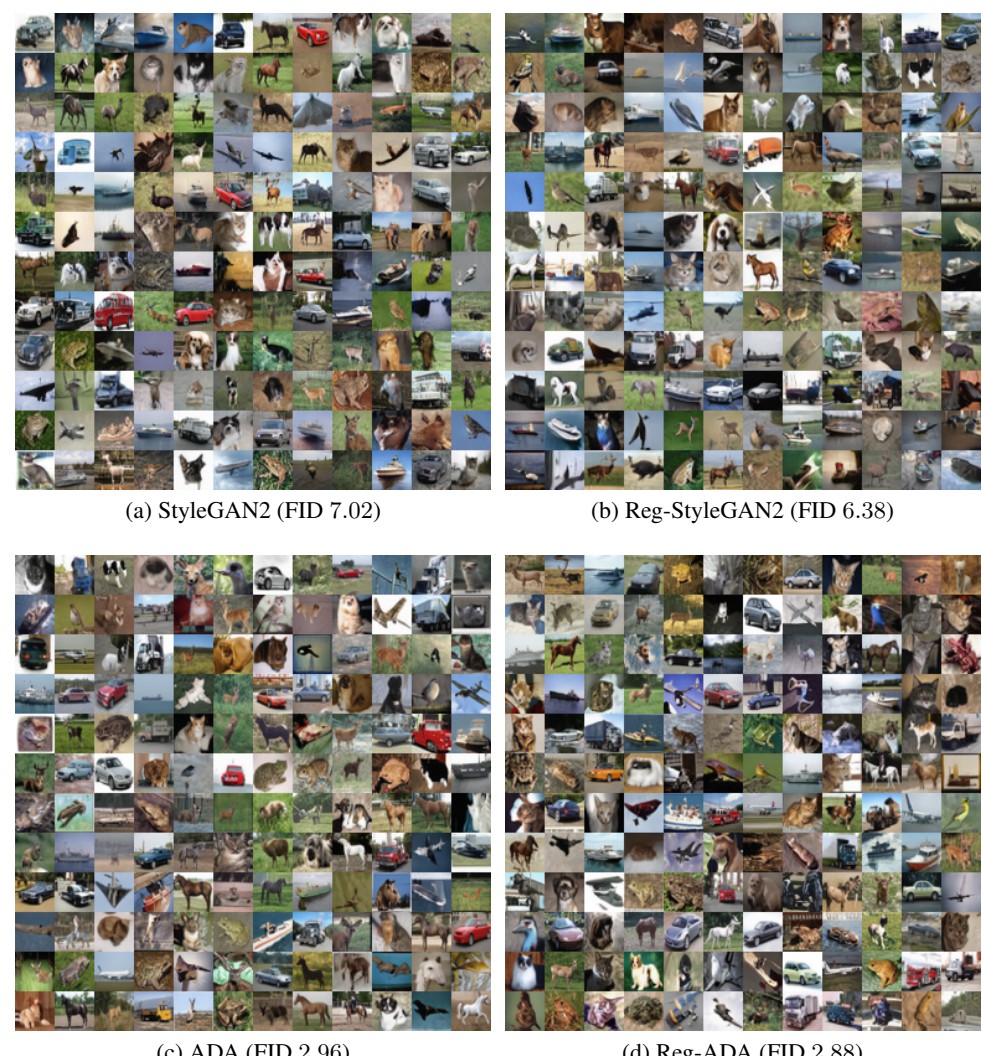

(a) StyleGAN2 (FID 7.02)

(b) Reg-StyleGAN2 (FID 6.38)

(c) ADA (FID 2.96)

(d) Reg-ADA (FID 2.88)

Figure 5: Samples generated for CIFAR-10, truncated ($\psi = 0.7$).

## C.3 LEARNING CURVES

We show the learning curves of GANs on FFHQ, LSUN CAT and CIFAR-10 in Fig. 6, Fig. 7 and Fig. 8 respectively. Reg-DGM consistently improves both baselines in all settings and the improvements increase as the number of the training data decreases. Moreover, the curves of Reg-DGM generally have a smaller fluctuation, which is consistent with our theory that the regularization reduces the variance of the baselines. One exception is Fig. 8 (a), which shows that Reg-DGM is more unstable than the baseline, which is caused by a bad random initialization. We mention that the instability of Reg-ADA in Fig. 6 (b) is due to that of ADA.

## C.4 ABLATION OF CLASSIFIER AND RESULTS UNDER MORE EVALUATION METRICS

In this section, to better understand the influence of classifiers on our method, we try to use pretrained classifiers with different regularization terms, layers, and backbones. We retain the same experimental setting as in Tab. 3.

**Regularization form.** We first investigate the feature matching objective (Salimans et al., 2016) as an alternative regularization term. Formally, it computes the square of the $l_2$-norm between expected features of real and fake samples from one layer of a feature extractor $f$, which can be represented

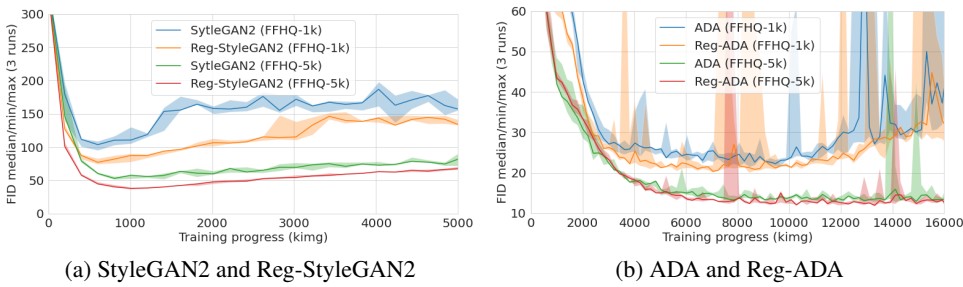

(a) StyleGAN2 and Reg-StyleGAN2    (b) ADA and Reg-ADA

Figure 6: Learning Curves on FFHQ.

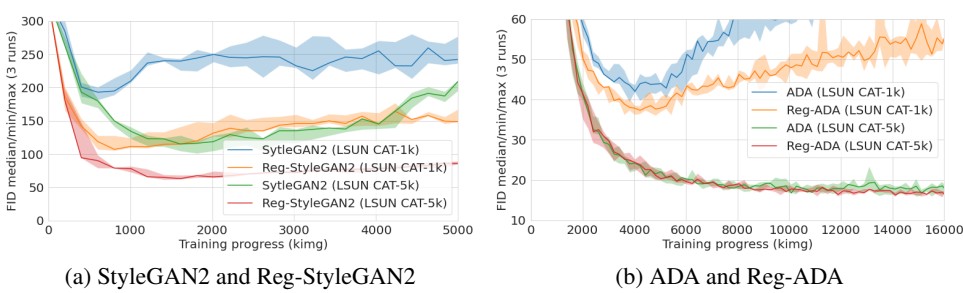

(a) StyleGAN2 and Reg-StyleGAN2    (b) ADA and Reg-ADA

Figure 7: Learning Curves on LSUN CAT.

as follows:

$$||\mathbb{E}_{x'\sim p_d}\left[f(x')\right] - \mathbb{E}_{x\sim p_g}\left[f(x)\right]||_2^2. \tag{36}$$

Note that the feature matching objective cannot be rewritten as the expectation over $p_g$ and thus cannot be understood as an energy function. As before, we adopt the last fully-connected layer of ResNet-18, and the results of StyleGAN2 regularized by Eq. (36) are shown in Tab. 6. Feature matching can greatly reduce FID of StyleGAN2 while it cannot improve the visual quality of the samples, as shown in Fig. 9.

Then, we evaluate the entropy-minimization regularization (Grandvalet & Bengio, 2004) as follows:

$$-\mathbb{H}(\text{softmax}(f(x))), \tag{37}$$

where $f(x)$ outputs the logits for the prediction distribution. As shown in Tab. 7, the entropy regularization achieves similar FID results to the baseline within a small search space of $\lambda$, showing the importance of the data dependency in the energy function.

**Layers in $f$.** To explore the different layers of a pre-trained model, we retrain GANs separately using the first convolution layer of ResNet-18 and the last layers of four modules in ResNet-18. As shown in Tab. 8, our method with features of diverse single layers can all improve the baseline StyleGAN2, and the last layer of ResNet-18 is most beneficial for our regularization strategy.

**Backbone of $f$.** We employ ResNet-50 and ResNet-101 as feature extractors to explore the effect of different backbones on Reg-DGM. Tab. 9 shows the results on FFHQ-5k. Even using the default $\lambda$ without tuning, Reg-DGM with ResNet-50 and ResNet-101 can achieve a similar FID to that of our default setting and outperform the baseline. We believe that Reg-DGM with ResNet-50 and ResNet-101 can get better results if we finely search the hyperparameter $\lambda$.

**Monte Carlo estimate of $f$.** We evaluate Reg-StyleGAN2 with 8, 16, 32, and 64 (the default batch size) samples to estimate the energy function in Eq. (5) of the main text, as shown in Tab. 10. We do not observe a significant improvement by increasing the number of samples. For instance, when $\lambda = 1$, the estimate with 64 samples achieves an FID of 38.10, which is similar to 37.77 of the single sample estimate. Intuitively, the features of faces are likely concentrated in a small area of the feature space of $f$, which is discriminative to other classes of natural images like cars, making the variance negligible to the training process of the generative model.

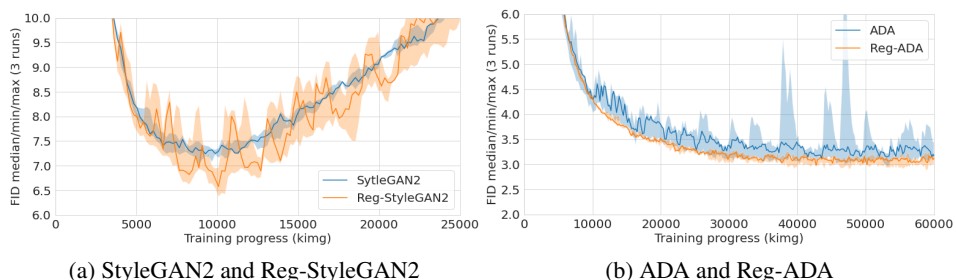

(a) StyleGAN2 and Reg-StyleGAN2        (b) ADA and Reg-ADA

Figure 8: Learning Curves on CIFAR-10.

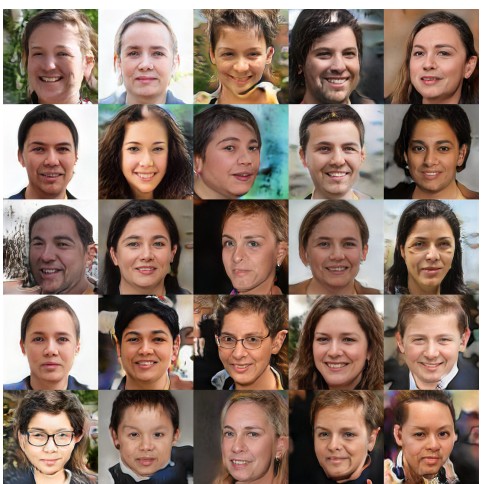

Figure 9: Samples generated for FFHQ-5k using feature matching (FID 32.65), truncated ($\psi = 0.7$).

**KID metric.** For a comprehensive comparison of Reg-DGM and baselines, we also evaluate the models under KID in Tab. 11. The conclusion remains the same as FID. Namely, Reg-DGM consistently improves baselines in all settings.

**Human evaluation.** We compare the quality of generated images from the human perspective by Amazon Mechanical Turk (AMT) as in prior work (Choi et al., 2020). In particular, we randomly generate $1,000$ pairs of images by StyleGAN2 and Reg-StyleGAN2 trained on FFHQ-5k using the truncation trick ($\psi = 0.7$). For each image pair, three unique workers in AMT choose the image with the more realistic and high-quality face image. Finally, $3,000$ selection tasks are completed by $34$ workers within $14$ hours. As consistent with the comparison of image quality between Fig. 4 (a) and Fig. 4 (b), Reg-StyleGAN2 is chosen in $63.5\%$ of the $3,000$ tasks, and other statistical information is shown in Tab. 12.

**Results on more training sets with different size.** To explore the influence of our method on training data with different sizes, we test the Reg-StyleGAN2 with the pre-trained ResNet-18 on different subsets of FFHQ. The results are shown in Tab. 13 and the hyperparameter $\lambda$ is simply fixed as 1 for new data settings 100, 2k, 7k, 10k, and 15k. Without heavily tuning the hyperparameter $\lambda$, Reg-StyleGAN2 shows consistent improvements over the baseline under the FID metric. There seems to be a trend that our method improves more on smaller datasets, which agrees with our Gaussian case illustrated by the Fig. 1 (b).

## C.5   SENSITIVITY ANALYSIS OF THE WEIGHTING HYPERPARAMETER

We empirically analyze the sensitivity of the weighting hyperparameter $\lambda$ on FFHQ-5k with Style-GAN2 as the base model in Tab. 14. It can be seen that $\lambda$ affects the performance significantly. Notably, although it is nearly impossible to get the optimal $\lambda$ via grid search, there is a range of $\lambda$ (e.g. from 0.01 to 1 in Tab. 14) such that Reg-DGM outperforms the base DGM, which agrees with

Table 6: The median FID ↓ on FFHQ and LSUN CAT with feature matching as the regularization. $\lambda$ is simply set as the same values in Tab. 3.

| Method | FFHQ | | LSUN CAT | |
|---|---|---|---|---|
| | 1k | 5k | 1k | 5k |
| StyleGAN2 (Karras et al., 2020b) | 103.66 | 52.71 | 186.55 | 115.16 |
| Reg-StyleGAN2 (**ours**) | **59.96** | **32.65** | **66.46** | **47.56** |

Table 7: Median FID on FFHQ-5k for the entropy-minimization regularization.

| $\lambda$ | (Baseline) | 1 | 0.1 | 0.01 | 0.001 | 0.0001 | 0.00001 |
|---|---|---|---|---|---|---|---|
| FID ↓ | 52.71 | 145.55 | 92.11 | 56.41 | 53.37 | 50.87 | 55.76 |

our theoretical analyses. Besides, Reg-DGM is not too sensitive when the value of $\lambda$ is around the optimum. For instance, the gap between the results of $\lambda = 0.1$ and $\lambda = 1$ in Tab. 14 is much smaller than their gain compared to the baseline. The performance of Reg-DGM deteriorates with a large $\lambda$ in Tab. 14 as expected. Proposition 2.2 shows that Reg-DGM is preferable if and only if its value is in a proper interval. Intuitively, a very large value means that we almost ignore the training data, which should lead to inferior performance.

Table 8: Results with different layers on FFHQ-5k. The "−1 layer" represents the last layer (i.e., our default setting). Note that layers are all indexed by the function *named_modules* in Pytorch. $\lambda$ is simply set as the same values from Tab. 3.

| Layer index | (Baseline) | 1 | 17 | 33 | 49 | 65 | −1 (by default) |
|---|---|---|---|---|---|---|---|
| FID ↓ | 52.71 | 46.39 | 47.65 | 51.61 | 45.24 | 46.01 | 37.77 |

Table 9: Results with different backbones on FFHQ-5k. $\lambda$ is simply set as the same values from Tab. 3.

| Backbone | (Baseline) | ResNet-18 | ResNet-50 | ResNet-101 |
|---|---|---|---|---|
| FID ↓ | 52.71 | 37.77 | 40.95 | 42.63 |

Table 10: FID ↓ on FFHQ-5k for different number of samples in MC.

| MC | 1 (by default) | 8 | 16 | 32 | 64 |
|---|---|---|---|---|---|
| $\lambda = 1$ | 37.77 | 40.97 | 40.78 | 37.54 | 38.10 |
| $\lambda = 10$ | 53.09 | 53.36 | 58.58 | 52.93 | 48.21 |

Table 11: KID$\times 10^3$ ↓ on FFHQ, LSUN-CAT, and CIFAR-10.

| Method | FFHQ | | LSUN CAT | | CIFAR-10 |
|---|---|---|---|---|---|
| | 1k | 5k | 1k | 5k | 50k |
| StyleGAN2 (Karras et al., 2020b) | 98.06 | 39.52 | 161.95 | 100.57 | $3.66 \pm 0.07$ |
| Reg-StyleGAN2 (**ours**) | **47.91** | **23.06** | **83.71** | **42.68** | $\mathbf{2.89 \pm 0.11}$ |
| ADA (Karras et al., 2020a) | 9.77 | 5.17 | 23.30 | 8.13 | $0.90 \pm 0.12$ |
| Reg-ADA (**ours**) | **9.38** | **4.30** | **20.52** | **6.54** | $0.83 \pm 0.06$ |

Table 12: Other statistical information about human evaluation. Number-$i$ indicates the number of image pairs that at least $i$ workers think StyleGAN2 or Reg-StyleGAN2 generates more realistic face images.

| Method | Number-1 | Number-2 | Number-3 |
|---|---|---|---|
| StyleGAN2 (Karras et al., 2020b) | 610 | 339 | 146 |
| Reg-StyleGAN2 (**ours**) | 854 | 661 | 390 |

Table 13: The FID on different subsets of FFHQ.

| Method | FFHQ | | | | | | |
|---|---|---|---|---|---|---|---|
| | 100 | 1k | 2k | 5k | 7k | 10k | 15k |
| StyleGAN2 | 132.68 | 103.66 | 83.32 | 52.71 | 37.75 | 34.20 | 23.07 |
| Reg-StyleGAN2 | **113.96** | **75.99** | **60.83** | **37.77** | **31.58** | **25.72** | **20.85** |

Table 14: Sensitivity analysis of $\lambda$ on FFHQ-5k with StyleGAN2 as the base DGM.

| Values of $\lambda$ | $\lambda = 0$ (base DGM) | $\lambda = 0.01$ | $\lambda = 0.1$ | $\lambda = 1$ | $\lambda = 10$ | $\lambda = 100$ |
|---|---|---|---|---|---|---|
| FID ↓ | 52.71 | 47.49 | 41.51 | 37.77 | 53.09 | 178.53 |

