# OpenReview forum: "Deep Generative Modeling on Limited Data with Regularization by Nontransferable Pre-trained Models"
_ICLR.cc/2023/Conference — ICLR 2023 poster_

### Official Review · Reviewer_5xKN · 2022-10-21

**Confidence:** 3
**Correctness:** 4
**Technical Novelty And Significance:** 3
**Empirical Novelty And Significance:** 3
**Recommendation:** 8

**Clarity, Quality, Novelty And Reproducibility:**

The paper is very clearly written, and the discussions and the experiments are nicely constructed to support the main claim of the paper. Also, the source code in the supplementary materials seems to be well documented.

**Strength And Weaknesses:**

### Strengths

- The paper is nicely written and easy to follow.
- The experimental results are clearly reported and support the claim well.
- The idea of using a non-generative model's features for defining an energy function is very interesting.

### Weaknesses

The analyses in Section 3 are not wrong, but the current way of presentation looks a little awkward. What's described here is that the facts that hold without the regularizer also hold with the regularizer under some conditions. In this sense, discussing those conditions in the main text, instead of deferring them to the appendix, would be more natural.

**Summary Of The Paper:**

**Potential double-blindness violation**

The source codes, scripts, and license files in the supplementary materials have the copyright notice of NVIDIA. It is not unnatural if only some of them have it since they may be third-party codes with the third-party being NVIDIA, but in this case, as far as I checked, all the files have it. I don't know if this would be regarded as a violation of double-blindness but report here just in case.

By the way, posting something independently from "Official Review" is not enabled at this stage, which I found a bit inconvenient for reporting this kind of thing.

---

**Summary of the paper**

The authors propose a regularization of learning generative models supposing the presence of an energy function that (hopefully) captures the true distribution's nature to some extent. Their theoretical discussion is twofold. One is about the fact that the regularization does not hurt the estimation as long as the hyperparameter is appropriately chosen and is shown for a prototypical Gaussian fitting example. Another is about the uniqueness of global optimum and probabilistic convergence of SGD to it. The authors show the benefits of the proposed regularization on different image datasets.

**Summary Of The Review:**

This is a solid paper. While the presentation around the analyses could possibly be improved, otherwise it is nicely written, and the experimental results well support the claim of the paper.

---

> ### Author Response · Authors · 2022-11-14
> **Response to Reviewer 5xKN**
>
> We thank reviewer 5xKN for the acknowledgement of our contributions and valuable suggestions.
>
> **Q1: The source codes, scripts, and license files in the supplementary materials have the copyright notice of NVIDIA.**
>
> Thank you for carefully checking our paper, appendices, and supplementary materials. As mentioned in Appendix B.2, our codes are based on the official codes of ADA (https://github.com/NVlabs/stylegan2-ada-pytorch) implemented by NVIDIA. Therefore, the source codes, scripts, and license files in the supplementary materials have the copyright notice of NVIDIA, which does not violate double-blindness.
>
> **Q2: The analyses in Section 3 are not wrong, but the current way of presentation looks a little awkward.**
>
> Thank you for checking our formulas and proofs carefully. We placed the descriptions of these conditions in Section 3.1  (Paragraph 4 on Page 4) of the revision to make the presentation clearer following your suggestion.

---

### Official Review · Reviewer_rg8D · 2022-10-24

**Confidence:** 3
**Correctness:** 4
**Technical Novelty And Significance:** 3
**Empirical Novelty And Significance:** 3
**Recommendation:** 6

**Clarity, Quality, Novelty And Reproducibility:**

Clarity - very good. Paper is well-written.
Quality - good. The experimental results seem sound. The theoretical analysis is a nice plus to help motivate the approach.
Novelty - ok. Feature-wise losses are used in many related tasks, but the authors try to motivate/justify their use here with some theoretical analysis. At first glance, the method seems similar to feature matching, which is why the discussion of the differences is very useful.
Reproducibility - seems straightforward to reproduce.

**Strength And Weaknesses:**

Strengths:

1) The paper is very well-written and easy to understand.
2) The authors present theoretical analysis to motivate/justify their approach.
2) The authors provide comprehensive experiences to demonstrate the effectiveness of their approach. The proposed regularization is applied to three models (StyleGAN, ADA, APA) trained for generation on three datasets (FFHQ, LSUN CAT, CIFAR-10), and provides a modest boost to each of the models. The authors' best model outperforms all the baselines in the main table.
4) The authors provide ablation studies and connect their proposed approach to other regularization strategies such as entropy minimization and feature matching.

Weaknesses:
1) There is a gap between the theoretical analysis and the actual method that is tested empirically. The bound of Section 2.1 is limited to the simple Gaussian example. The theoretical results do not provide intuition/justification for using a pretrained feature extractor.
2) The comparison between the proposed approach and existing regularization strategies such as entropy minimization and feature matching should be emphasized, both in discussion and in results. At the moment, the results for the other two regularization strategies are in the Appendix and the presentation of the results does not easily allow the reader to compare them to the proposed approach. For example, it seems that feature matching achieves better FID than the proposed method, but this is not clear from the presentation.

**Summary Of The Paper:**

This paper presents a regularization strategy for training deep generative models such as GANs on limited data. The strategy involves adding an additional term to the objective function for the generator that penalizes the expected mean square error between the features of a generated sample and a real sample, where the features are extracted from a pre-trained model. The authors present some theoretical analysis to motivate their approach and provide comprehensive empirical results to demonstrate its effectiveness.

**Summary Of The Review:**

The method of regularization (expected feature-wise MSE) is not a novel concept, but the authors motivate their use here with some theoretical analysis and show promising experimental results. I would suggest more clearly discussing the differences between this work and other regularization strategies such as feature matching and entropy minimization, as well as showing the quantitative comparison in the main paper.

---

> ### Author Response · Authors · 2022-11-14
> **Response to Reviewer rg8D**
>
> We thank reviewer rg8D for the acknowledgement of our contributions and the valuable comments.
>
> **Q1: There is a gap between the theoretical analysis and the actual method that is tested empirically. The bound of Section 2.1 is limited to the simple Gaussian example. The theoretical results do not provide intuition/justification for using a pretrained feature extractor.**
>
> Thanks for the valuable comment. In the Gaussian example, we show that the gain of Reg-DGM gets larger (namely, MSE[Reg-DGM] / MSE[MLE] gets smaller) as the pre-trained model is closer to the target data distribution (proof in Appendix A.1.1, illustration in Figure 1 (c)). The theorem does not directly justify the usage of a pre-trained feature extractor but the same intuition holds: if the feature extractor is pre-trained on a related dataset, then it can induce an EBM that is not too far from the target data distribution and can be beneficial.
>
> **Q2: The comparison between the proposed approach and existing regularization strategies such as entropy minimization and feature matching should be emphasized, both in discussion and in results.**
>
> Thanks for the valuable comment and suggestion. We moved the quantitative results and discussion of the existing regularization strategies in Appendix C.4 to Section 5.3 (Paragraph 6 on Page 8) in the revision to make it clearer.
>
> Nevertheless, our main contribution is the general framework instead of a specific energy function. We did not use the feature matching loss in Reg-DGM only because it did not produce samples of better visual quality than the baseline.

---

### Official Review · Reviewer_ZXuR · 2022-10-24

**Confidence:** 4
**Correctness:** 2
**Technical Novelty And Significance:** 3
**Empirical Novelty And Significance:** 2
**Recommendation:** 6

**Clarity, Quality, Novelty And Reproducibility:**

Clarity: The presentation is generally clear.
Quality: The quality of this paper is fair.
Novelty: The idea is simple and intuitive, but has limited technical novelty.
Reproducibility: Code for some experiments is attached.

**Strength And Weaknesses:**

Strength:
The paper contributes some new ideas. It provides an alternative way compared to fine-tuning-based methods to help generative modeling in limited data. Experiments with Reg-StyleGAN2, Reg-ADA, and Reg-ADA-APA show the proposed method outperforms the corresponding base models.

Weaknesses:
It is not clear why incorporating pre-trained models as regularization can improve generative modeling. It is said that this can reduce variance. However, it will also introduce bias. Does any pretrained models always help?

Furthermore, it is claimed in the paper that the paper focuses on the limited data setting. However, I don't see any data settings described in the experiments. In the appendix, it has some descriptions on how to split the training data, but it is too rough to know whether they use the original whole dataset? If so, I am not sure if it is on the setting of "limited data". In my opinion, limited data means few-shots. So it should at least have some studies on the performance w.r.t. the number of training data.

In addition, to show the effectiveness of the proposed method, existing methods to deal with limited data should be compared, e.g., the methods discussed in the Related Work. The current results do not have these comparisons, thus it is difficult to claim the proposed method is a better choice.


**Summary Of The Paper:**

To address the limited data in generative modeling, this paper proposes a regularized deep generative model (Reg-DGM) to reduce the variance and prevent over-fitting. There are two main contributions of this paper: 1) proposed a regularized deep generative model based on pre-trained model with an energy function. 2) Experiments on three main datasets and serval base model demonstrate that RegDGM improves the generation performance and outperform the SOTA methods.

**Summary Of The Review:**

This paper proposes a simple regularized deep generative model by leveraging pretrained models to deal with limited training data. The experiments demonstrate the effectiveness of the method on some datasets compared with the unregularized versions. However, the motivation is not well demonstrated, and not enough evidence is shown to demonstrated the effectiveness of the proposed method.

Post rebuttal:
Thanks for the clarifications. Most of my concerns are addressed, and I am happy to raise my score to 6. The reason of not raising to higher scores is that I think the proposed method is somewhat restricted. I think it improves the generative model only if the distribution of the pretrained models has a distribution similar enough to the training data, which is hard to guarantee in practice. In addition, the paper only shows successful cases, it would be great to see some failure cases to see how much worse a bad-pretrained model can bring to the generative model.

---

> ### Author Response · Authors · 2022-11-14
> **Response to Reviewer ZXuR (Q1-Q3)**
>
> We thank reviewer ZXuR for the great efforts and the valuable comments.
>
> **Q1: It is not clear why incorporating pre-trained models as regularization can improve generative modeling. It is said that this can reduce variance. However, it will also introduce bias. Does any pre-trained model always help?**
>
> Thanks for the valuable comment. In a sentence, any pre-trained model will not hurt the performance, and it significantly helps if the corresponding EBM is close to the target distribution. We explain it in detail below.
>
> Intuitively, training a model on limited data suffers from a large variance (overfitting) and using a pre-trained model suffers from a large bias (underfitting). Thus, Reg-DGM with a proper hyperparameter can balance between them and achieve a better MSE, a common measure in parameter estimation that considers both bias and variance.
>
> Theoretically, in the Gaussian case, we prove that, with a proper $\lambda$, any pre-trained model always helps (Proposition 2.2). The gain gets larger (namely, MSE[Reg-DGM] / MSE[MLE] gets smaller) as the pre-trained model is closer to the target data distribution (proof in Appendix A.1.1, illustration in Figure 1 (c)).
>
> Such an intuition also holds in the deep learning case: the possibility to choose a small enough $\lambda$ ensures that the pre-trained model does not hurt the performance. However, to obtain a significant improvement, we require that the pre-trained model should be close to the target data distribution.
>
> Empirically, we employ three representative types of pre-trained models, including ResNet trained on ImageNet, CLIP trained on text-image pairs, and FaceNet trained on face images. All pre-trained models consistently and significantly improve baselines on datasets related to the pre-training data and outperform a large family of strong competitors, which supports our theoretical insights.
>
> We added the above discussion in Section 2.1 (Paragraph 2 on Page 4) in the revision.
>
> **Q2: Data settings are not described in the experiments. In the appendix, it has some descriptions on how to split the training data, but it is too rough to know whether they use the original whole dataset?**
>
> Thanks for the valuable comment. We **do NOT** use the original whole dataset on FFHQ and LSUN CAT, as detailed below.
>
> For a fair comparison to a large family of prior work, our data processing is the same as that of ADA (Karras et al., 2020a), followed by DA (Zhao et al., 2020a), InsGen (Yang et al., 2021), GenCo (Cui et al., 2021), APA (Jiang et al., 2021), which are representative baselines.
>
> In particular, the training set is of size 1k on FFHQ and 5k on LSUN CAT. We randomly split the training subsets on FFHQ and LSUN CAT in the same way as ADA (Karras et al., 2020a). For reproducibility, we directly use the random seed provided by the official implementation of ADA in https://github.com/NVlabs/stylegan2-ada-pytorch.
>
> We added these details in both Section 5 (Paragraph 6 on Page 6) and Appendix B.2 (Paragraph 9 on Page 18) of the revision.
>
> **Q3: Are the used data settings on the setting of "limited data"? Add some studies on the performance w.r.t. the number of training data.**
>
> Thanks for the valuable comment. We clarify that we refer to our settings as **"limited data setting" following the prior work** including ADA (Karras et al., 2020a), DA (Zhao et al., 2020a), $R_{LC}$
>  (Tseng et al., 2021), GenCo (Cui et al., 2021) and APA (Jiang et al., 2021).
>
> To explore the influence of our method on training data with different sizes, we provide ablation experiments on FFHQ with training sets of sizes 100, 1k, 2k, 5k, 7k, 10k, and 15k. The following table shows FID on different subsets of FFHQ and the hyperparameter $\lambda$ is simply fixed as $1$ for Reg-StyleGAN2 on new data settings, (i.e. 100, 2k, 7k, 10k, and 15k). Without heavily tuning the hyperparameter $\lambda$, Reg-StyleGAN2 shows consistent improvements over the baseline under the FID metric. There seems to be a trend that our method improves more on smaller datasets, which agrees with our Gaussian case (illustration in Figure 1 (b)). We put these experiments and results in Appendix C.4 (Paragraph 2 on Page 24) in the revision.
>
> |Method (FID $\downarrow$)| 100|1000| 2000| 5000| 7000| 10000| 15000|
> | :----: | :----: | :----: | :----: | :----: |:----: |:----: |:----: |
> |StyleGAN2 |132.68| 103.66|83.32 | 52.71|37.75| 34.20| 23.07|
> |Reg-StyleGAN2|113.96| 75.99| 60.83 | 37.77| 31.58 | 25.72 |20.85|
> ||||||
>
> Thanks again for the constructive comment, and we believe this can improve the quality of our article.

---

> > ### Author Response · Authors · 2022-11-14
> > **Response to Reviewer ZXuR (Q4)**
> >
> > **Q4: To show the effectiveness of the proposed method, existing methods to deal with limited data should be compared, e.g., the methods discussed in the Related Work.**
> >
> > Thanks for the valuable comment. In Table 1 of the main text, we directly compare our method with 11 representative baselines out of 14 existing methods mentioned in the Related Work section. In particular, we compare our method with the current SOTA method ADA+APA (Jiang et al., 2021; Li et al., 2022), the representative fine-tuning approaches (Wang et al., 2018; Mo et al., 2020), and the most direct and popular baselines StyleGAN2 (Karras et al., 2020b) and ADA (Karras et al., 2020a). Therefore, it is sufficient to show the effectiveness of our method.
> >
> > There are three fine-tuning methods (Wang et al., 2020; Li et al., 2020; Ojha et al., 2021) not listed in Table 1. In fact, they are not directly comparable due to different experimental settings (such as data settings). Nevertheless, our contribution is orthogonal to them.
> >
> >
> > **Thanks again for your comment. We believe that the feedback addresses your concerns and we are happy to address any further one.**

---

> > > ### Author Response · Authors · 2022-11-17
> > > **Looking forward for your reply**
> > >
> > > Dear reviewer ZXuR,
> > >
> > > Thanks again for your valuable comments. In the rebuttal (as well as the revision), we clarified the bias-variance trade-off issue, the limited data setting, and related work that are not directly comparable. We also added new experiments to show the performance of Reg-DGM with different numbers of training samples.
> > >
> > > We believe that our rebuttal clarified all the concerns and we are happy to reply to further questions if any.

---

> > > > ### Author Response · Authors · 2022-11-30
> > > > **Sincerely looking forward to the further discussions**
> > > >
> > > > Dear reviewer ZXuR,
> > > >
> > > > We are wondering if our response and revision have resolved your concerns. If our response has addressed your concerns, we would highly appreciate it if you could re-evaluate our work and consider raising the score.
> > > >
> > > > If you have any additional questions or suggestions, we would be happy to have further discussions.
> > > >
> > > > Best regards,
> > > >
> > > > The Authors

---

> > > > > ### Author Response · Authors · 2022-12-08
> > > > > **Last post**
> > > > >
> > > > > Dear reviewer ZXuR,
> > > > >
> > > > > Thanks for your comments again. We provided point-to-point feedback to your questions and polished the paper following your suggestions. Since we posted our rebuttal on 16 Nov, we received no further comment. Therefore, we believe that your concerns are all addressed. Please re-evaluate our work and consider raising the score in the final decision.

---

> > > > > > ### Comment · Reviewer_ZXuR · 2022-12-11
> > > > > > **sorry for the delay**
> > > > > >
> > > > > > Thanks for the rebuttal. Most of my concerns have been addressed, and I have raised my score to 6. Please see my updated review for more details.

---

> > > > > > > ### Author Response · Authors · 2022-12-11
> > > > > > > **Thank you**
> > > > > > >
> > > > > > > Dear reviewer ZXuR,
> > > > > > >
> > > > > > > We sincerely appreciate your reply. Thanks again for your valuable comment to improve the quality of the paper.
> > > > > > >
> > > > > > > Best,
> > > > > > > Authors.

---

### Official Review · Reviewer_rAFU · 2022-10-30

**Confidence:** 3
**Clarity, Quality, Novelty And Reproducibility:** Please see above comments.
**Correctness:** 3
**Technical Novelty And Significance:** 2
**Empirical Novelty And Significance:** 2
**Recommendation:** 6

**Strength And Weaknesses:**

- This paper is generally well-written and the overall structure is clear. The motivation is clearly stated.
- The authors suggest looking at the problem and proposed model from the bias-variance trade-off perspective. However, the presented theoretical analysis does not seem to be very intuitive in explaining why and how this perspective would help. To put it in another way, bringing another pre-trained model from a larger-scale dataset into the training of a generative model under a limited data regime is not a very surprising idea in the first place, which makes the synthetic study and the theoretical analysis a bit obsolete.
- Why does the author emphasize the role of an energy-based formulation of $f$ rather than a regularization term, which is pretty much a perceptual loss in previous works? What does the distribution induced by the EBM imply in the context of the proposed model and its training process?
- The exact description of how the limited data is prepared is not very clear, as well as how the metrics are computed. For example, are the FID scores computed against the original whole training set or in the limited dataset?


**Summary Of The Paper:**

This work studies generative modeling under a limited data regime and proposes a regularized generative model by utilizing a pre-trained model from an external dataset. The paper presented some theoretical justification and analysis of the proposed approach from the bias-variance trade-off perspective. Experiments on image generation are conducted in comparison to several baseline methods, using GANs as the backbone generative model, showing that the proposed model can achieve better performance with the help of the proposed regularization approach.

**Summary Of The Review:**

This work studies the generative models under a limited data regime which is interesting, some parts including theoretical analysis and evaluation can be improved to make the contributions clearer in terms of soundness and novelty.

---

> ### Author Response · Authors · 2022-11-14
> **Response to Reviewer rAFU**
>
> We thank reviewer rAFU for the acknowledgement of our contributions and the valuable comments.
>
> **Q1: The presented theoretical analysis does not seem to be very intuitive in explaining why and how this perspective would help.**
>
> Thanks for the valuable comment. Intuitively, training a model on limited data suffers from a large variance (overfitting) and using a pre-trained model suffers from a large bias (underfitting). Thus, our method with a proper hyperparameter can balance between them and achieve a better MSE and expected risk, which are commonly used metrics in statistics and machine learning theory. Such an intuition holds in both the Gaussian case and the deep learning case. As discussed in Section 2.1, we conduct the Gaussian case to quantify the intuition formally (Proposition 2.2) while such an analysis is extremely challenging and generally open in deep learning.
>
> We made the intuition of the Gaussian case clearer in Section 2.1 (Paragraph 2 on Page 4) in the revision.
>
> **Q2: Why does the author emphasize the role of an energy-based formulation of $f$ rather than a regularization term? What does the distribution induced by the EBM imply?**
>
> Thanks for the valuable question. We clarify the role of the energy-based formulation here. Although both formulations result in the same training process, the energy-based formulation provides a complementary perspective. Theoretically, it directly inspires our analysis in the Gaussian example, where we can treat the regularization as a divergence between two Gaussians to obtain analytic solutions immediately. Intuitively, it provides an alternative way to understand our method: it encourages the model to produce samples with a high likelihood evaluated by a pre-trained EBM.
>
> **Q3: The exact description of how the limited data is prepared is not very clear, as well as how the metrics are computed.**
>
> Thanks for the valuable comment. Due to a strict page limit, we deferred the detailed descriptions to Appendix B.2 in the submission.
>
> For a fair comparison to a large family of prior work, our data processing and metric calculating are the same as those of ADA (Karras et al., 2020a), followed by DA (Zhao et al., 2020a), InsGen (Yang et al., 2021), GenCo (Cui et al., 2021), APA (Jiang et al., 2021), which are representative baselines.
>
> In particular, we use 1k and 5k subsets of FFHQ and LSUN CAT, and full CIFAR-10 as training sets. We randomly split training subsets of size 1k and 5k from full FFHQ and LSUN CAT in the same way as ADA (Karras et al., 2020a). For reproducibility, we directly use the random seed provided by the official implementation of ADA in https://github.com/NVlabs/stylegan2-ada-pytorch.
>
> Following the direct and strong baseline ADA, we compute the FID and KID between 50, 000 generated images and **the original whole training set** instead of the limited dataset, and we report the medium FID on FFHQ and LSUN CAT and the mean FID with standard deviation on CIFAR-$10$ out of $3$ runs. We record the best FID during training in each run as suggested by ADA.
>
> We added these details in both Section 5 (Paragraph 6 on Page 6) and Appendix B.2 (Paragraph 9 on Page 18) of the revision.

---

### Decision · Program_Chairs · 2023-01-20

**Decision:**

Accept: poster

**Justification For Why Not Higher Score:**

For a higher score there should be a stronger link between theory and practice.

**Justification For Why Not Lower Score:**

Their could a lower score (reject) with a strong motivation to improve the outstanding deficiencies. However, the paper is good enough as it stands.

**Metareview: Summary, Strengths And Weaknesses:**

This paper is about using a pre-trained model for regularization when learning a generative model from limited data. The paper analyses theoretically the Gaussian case and then goes on to demonstrate the approach in deep learning settings.

The reviewers appreciate the contributions with a few reservations regarding the connect between the theoretical and practical scenarios and the conclusiveness of the empirical work.

Still the overall attitude support acceptance.

**Note From Pc:**

if the above contains the word "oral" or "spotlight" please see: "oral" presentation means -> notable-top-5% and "spotlight" means -> notable-top-25%. As stated in our emails, we are disassociating presentation type from AC recommendations